https://doi.org/10.1038/s41467-019-11002-5　　**OPEN**

# Iron-dependent histone 3 lysine 9 demethylation controls B cell proliferation and humoral immune responses

Yuhang Jiang [1,2,12], Cuifeng Li[1,2,12], Qian Wu [3,4,5,12], Peng An[3,4,5], Laiquan Huang [6], Jia Wang[3,4,5], Chen Chen[7], Xi Chen[1,2], Fan Zhang[3,4,5], Li Ma[8], Sanhong Liu[2], Hanqing He[9], Shuyun Xie[9], Yangbai Sun[9], Hanshao Liu[1,2], Yu Zhan[2], Yu Tao[2], Zhi Liu[1,2], Xiaohua Sun[1,2], Yiming Hu[1,2], Qi Wang[2], Deji Ye[2], Jie Zhang[2], Shanhua Zou[7], Ying Wang[2], Gang Wei[8], Yongzhong Liu [10], Yufang Shi [2], Y. Eugene Chin[2], Yongqiang Hao[11], Fudi Wang [3,4,5] & Xiaoren Zhang[1,2]

Trace elements play important roles in human health, but little is known about their functions in humoral immunity. Here, we show an important role for iron in inducing cyclin E and B cell proliferation. We find that iron-deficient individuals exhibit a significantly reduced antibody response to the measles vaccine when compared to iron-normal controls. Mice with iron deficiency also exhibit attenuated T-dependent or T-independent antigen-specific antibody responses. We show that iron is essential for B cell proliferation; both iron deficiency and α-ketoglutarate inhibition could suppress cyclin E1 induction and S phase entry of B cells upon activation. Finally, we demonstrate that three demethylases, KDM2B, KDM3B and KDM4C, are responsible for histone 3 lysine 9 (H3K9) demethylation at the cyclin E1 promoter, cyclin E1 induction and B cell proliferation. Thus, our data reveal a crucial role of H3K9 demethylation in B cell proliferation, and the importance of iron in humoral immunity.

[1] Affiliated Cancer Hospital & Institute, Guangzhou Medical University, 510000 Guangzhou, China. [2] CAS Key Laboratory of Tissue Microenvironment and Tumor, Shanghai Institute of Nutrition and Health, Shanghai Institutes for Biological Sciences, University of Chinese Academy of Sciences, Chinese Academy of Sciences, 200031 Shanghai, China. [3] The First Affiliated Hospital, School of Public Health, Institute of Translational Medicine, Zhejiang University School of Medicine, 310058 Hangzhou, China. [4] Beijing Advanced Innovation Center for Food Nutrition and Human Health, China Agricultural University, 100193 Beijing, China. [5] Department of Nutrition, Precision Nutrition Innovation Center, School of Public Health, Zhengzhou University, 450001 Zhengzhou, China. [6] Department of Hematology, The First Affiliated Hospital of Wannan Medical College, 241000 Wuhu, China. [7] Department of Hematology, Zhongshan Hospital Fudan University, 200032 Shanghai, China. [8] Key Laboratory of Computational Biology, Chinese Academy of Sciences-Max Planck Partner Institute for Computational Biology, Shanghai Institute of Nutrition and Health, University of Chinese Academy of Sciences, Chinese Academy of Sciences, 200031 Shanghai, China. [9] Zhejiang Provincial Center for Disease Control and Prevention, 310051 Hangzhou, China. [10] Shanghai Cancer Institute, Shanghai Jiao-Tong University School of Medicine, 200240 Shanghai, China. [11] Shanghai Key Laboratory of Orthopaedic Implants, Department of Orthopaedics, Shanghai Ninth People's Hospital, Shanghai JiaoTong University School of Medicine, 200011 Shanghai, China. [12] These authors contributed equally: Yuhang Jiang, Cuifeng Li, Qian Wu. Correspondence and requests for materials should be addressed to F.W. (email: fwang@zju.edu.cn) or to X.Z. (email: xrzhang@sibs.ac.cn)

The trace element iron is essential for many fundamental metabolic processes and biochemical activities in cells and organisms[1]. In eukaryotes, extracellular circulating $Fe^{3+}$-transferrin (Tf) complex binds to the transferrin receptor, $Fe^{3+}$ should be reduced by STEAP3 to $Fe^2$, consequently $Fe^2$ is transported from endosome into the cytoplasm for utilization[2–4]. A missense mutation in TFRC, which encodes Tf receptor 1 (TFR1), has been reported to be associated with impaired T and B lymphocyte development or function[5]. However, the current understanding of the role of iron in adaptive immunity remains limited and is, in some ways, controversial[6–9].

Within the immune system, immature B cells in bone marrow migrate to the spleen and differentiate into mature B cells, including marginal zone B (MZB) cells and follicular B (FOB) cells. Differentiation of resting B cells into plasma cells and class switching are known to be dependent on the normal activation and rapid proliferation of B cells[10,11]. Mature B cells express both B-cell antigen receptors (BCRs) and toll-like receptors (TLRs) on their surface. MZB cells primarily participate in T-cell-independent (TI) immune responses, and resting mature B cells stimulated with TLR ligands (such as lipopolysaccharides [LPS]) are used extensively as TI immune response models in vitro[12]. In contrast, FOB cells are primarily responsible for T-cell-dependent (TD) immune responses, during which BCR-activated FOB cells contribute greatly to germinal center formation, class-switch recombination, and antibody responses[13,14].

Histone lysine methylation has been reported to play important roles in regulating a wide range of processes through transcriptional activation or repression[15,16]. In germinal center B (GCB) cells, EZH2 is highly expressed and mediates the transcriptional repression of several tumor suppressor genes by catalyzing their H3K27 trimethylation[17]. However, it remains unknown whether and how histone modification regulates B-cell activation, proliferation, and differentiation.

In the present study, we demonstrate that iron-dependent H3K9 demethylation is essential for cyclin E1 induction and B-cell proliferation in response to BCR or TLR stimulation. Iron deficiency in mice led to dramatically attenuated TD and TI antigen-specific antibody responses, and human patients with iron deficiency presented significantly weakened antibody responses to the measles vaccine (MV). In iron-deficient or α-ketoglutarate (2-OG)-inhibited B cells, cyclin E1 induction and S phase entry during B-cell proliferation driven by BCR or TLR stimulation were impaired due to inhibited H3K9me demethylation at the promoter region of cyclin E1, resulting in abnormal S phase entry. Our data reveal a critical function of iron-dependent H3K9 demethylation in B-cell proliferation and establish a link between iron and humoral immunity.

## Results

### The iron levels in serum correlate with antibody responses to vaccination in humans.

First, we sought to determine whether humoral immunity was influenced by abnormal iron metabolism. We conducted a clinical investigation at the Zhejiang Provincial Center for Disease Control and Prevention to assess the correlation between iron metabolism and antibody responses to the MV in humans. MV-specific immunoglobulin (Ig) G antibody titers, serum iron levels, and Tf saturation (TS) were measured. Interestingly, among 118 individuals aged 10 years and older, MV-specific IgG antibody titers in individuals with iron deficiency (serum iron < 50 μg/dl) were markedly lower than those in individuals with normal iron levels (serum iron ≥ 50 μg/dl). Consistent results were obtained for transferrin saturation (individuals with TS < 16% were typically defined as having iron deficiency; Fig. 1a, b). Individuals with a low antibody response

(MV antibody titer below 200 mIU/ml) showed significantly lower levels of serum iron than those with a normal (MV antibody titer between 200 and 800 mIU/ml) or high antibody response (MV antibody titer >800 mIU/ml), similar results were obtained for transferrin saturation (Fig. 1c). Correlation analysis revealed that there were significant and positive correlations between MV-specific IgG antibody titers and serum iron levels and between MV-specific antibody titers and transferrin saturation (Fig. 1d). Meanwhile, some other factors that might be involved (including other trace elements and Vitamin D) were also measured in this cohort. However, results revealed that in population 10 years and above, no obvious correlation between these factors and MV-specific antibody titers were observed (Supplementary Fig. 1a). These data indicated a close correlation between iron and humoral immune responses in humans.

### Altered immunoglobulin responses in iron-deficient mice.

To address whether a deficient supply of iron would result in altered humoral immune responses in mice, we established an iron-deficient mouse model using iron-deficient diet. Compared with control mice, serum iron in mice fed with an iron-deficient diet was sharply decreased (Supplementary Figs. 2a and 3a). In iron-deficient mice, mature B-cell population was significantly reduced in the periphery, and circulating mature B cells in bone marrow were markedly reduced, but the immature B cell population in bone marrow was normal (Fig. 2a, b and Supplementary Fig. 3b). In addition, no significant difference was detected in the proportion of splenic follicular B cells and marginal zone B cells between iron-deficient mice and control mice, indicating that B-cell development and maturation were not impaired in the iron-deficient mice (Supplementary Fig. 3b, c).

Next, we sought to assess the in vivo antibody responses of iron-deficient mice. When basal immunoglobulin levels in unimmunized mice were measured, basal level of various serum IgG isotypes did not change significantly, but results indicated that IgM level was lower in iron-deficient mice than in control mice (Supplementary Fig. 3d). Intriguingly, after mice were immunized with two types of TI antigens that primarily lead to IgM and IgG3 antibody responses, induced secretion of antigen-specific IgG3 and IgM was severely attenuated in iron-deficient mice compared with control mice regardless of whether the mice underwent immunization with 2,4-dinitrophenol (DNP)-Ficoll (TI-2) or 2,4,6-trinitrophenyl (TNP)-LPS (TI-1) (Fig. 2c, d).

When we immunized the two groups of mice with a TD antigen, DNP-keyhole limpet hemocyanin (KLH), mice with iron deficiency showed a dramatic reduction in antigen-specific IgG1 and IgM production vs. the mice with adequate iron (Fig. 2e). Mice were re-immunized with DNP-KLH for a second time at day 21, which lead to a strong recall immune response at day 28 in normal mice; but in iron-deficiency mice, the recall immune response was weak (Fig. 2e). Furthermore, in order to follow the antibody titers in iron-deficient mice vs. normal mice after long-term immunization, we collected eyelid blood samples and examined the antibody titers 2 months after the DNP-KLH immunization mice. Results showed significantly lower levels of antigen-specific IgG1 and IgM after long-term immunization in iron-deficient mice compared to control mice (Fig. 2e).

The reduction in IgG production suggested that germinal center formation, a necessary process for the generation of high-affinity IgG1 antibodies, might have been attenuated in the iron-deficient mice. Ten days after immunization with sheep red blood cells (SRBCs) as polyclonal antigens, control mice displayed an expected increase in splenic germinal center B cells (defined as $B220^+IgD^{lo}Fas^+GL7^+$; (Supplementary Fig. 2b)). In contrast, the splenic germinal center B cells in iron-deficient mice were

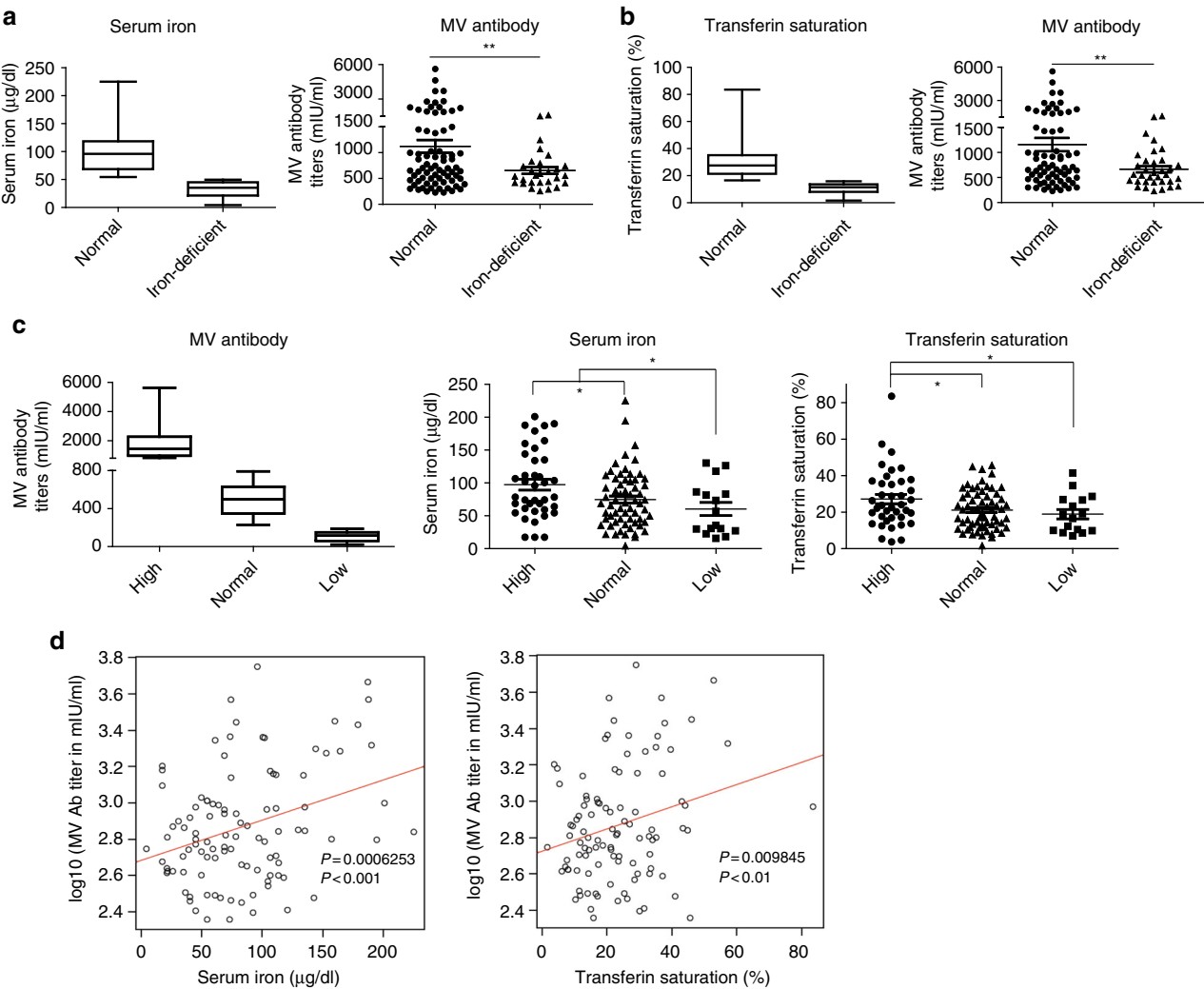

**Fig. 1** MV response is positively correlated with iron metabolism indicators in humans. **a**, **b** MV-specific IgG antibody titers in human patients with iron deficiency (defined as serum iron < 50 μg/dl or TS < 16%) vs. normal subjects. *P < 0.05, **P < 0.01, Student's t-test. **c** Serum iron levels and transferrin saturation in individuals with different degrees of MV response. *P < 0.05, **P < 0.01, Student's t-test. **d** A linear regression model was used to analyze the relationship between serum iron parameters (serum iron and TS) and MV concentrations in humans. The scatter-plot graphs with error bars indicate the mean ± SEM, and the box and whisker graphs with error bars indicate the range from the minimum value to the maximum value

significantly reduced (Fig. 2f, g). Consistently, histological analysis of the spleens from mice immunized with SRBCs showed that fewer germinal center regions were present in iron-deficient mice than in iron-normal mice (Fig. 2h).

Collectively, iron deficiency reduced the mature B-cell population, inhibited antigen-specific Ig production, and impaired germinal center formation, suggesting that iron might play an important role in B-cell functions and humoral immune responses.

**Defects in mature B-cell population and B-cell proliferation in Steap3-KO mice.** The metalloreductase STEAP3 is required for iron uptake in eukaryotes[4]. We thus further investigated the influence of iron uptake on humoral immune responses with regards to STEAP3. STEAP3 was preferentially expressed in splenic B cells rather than in CD4+/CD8+ T cells and dendritic cells (Supplementary Fig. 3a); in addition, STEAP3 expression in B cells was much higher than that of the other three STEAP family members (Supplementary Fig. 3b).

Similar to iron-deficient mice, Steap3-knockout (KO) mice (Supplementary Fig. 2c) displayed reductions in mature B-cell populations in the spleen and in the number of circulating mature B cells in the bone marrow (Fig. 3a, b and Supplementary Fig. 4d). In addition, no difference in the frequency of splenic FOB cells or MZB cells was observed between Steap3-KO mice and wild-type mice, indicating that B-cell development and maturation were not altered in Steap3-KO mice (Supplementary Fig. 3c).

Next, we measured the ability of Steap3-KO B cells to proliferate in response to BCR, TLR, and CD40 stimulation. As assessed by tritiated thymidine incorporation and cell viability assays, Steap3-KO B cells proliferated poorly compared with wild-type B cells in response to anti-IgM, LPS, and anti-CD40 stimulation (Fig. 3c). We further checked in vivo antibody responses of Steap3-KO mice immunized with T-cell-independent antigen TNP-LPS, results showed that compared with wild-type mice, TI antigen-induced secretion of antigen-specific IgG3 and IgM was severely attenuated in Steap3-KO mice (Fig. 3d).

Steap3-KO mice showed signs of anemia. To investigate whether the B-cell defects caused by the deletion of Steap3 were intrinsic to B cells, we generated bone marrow chimeric mice. Six weeks later after bone marrow transfer, in mice that received

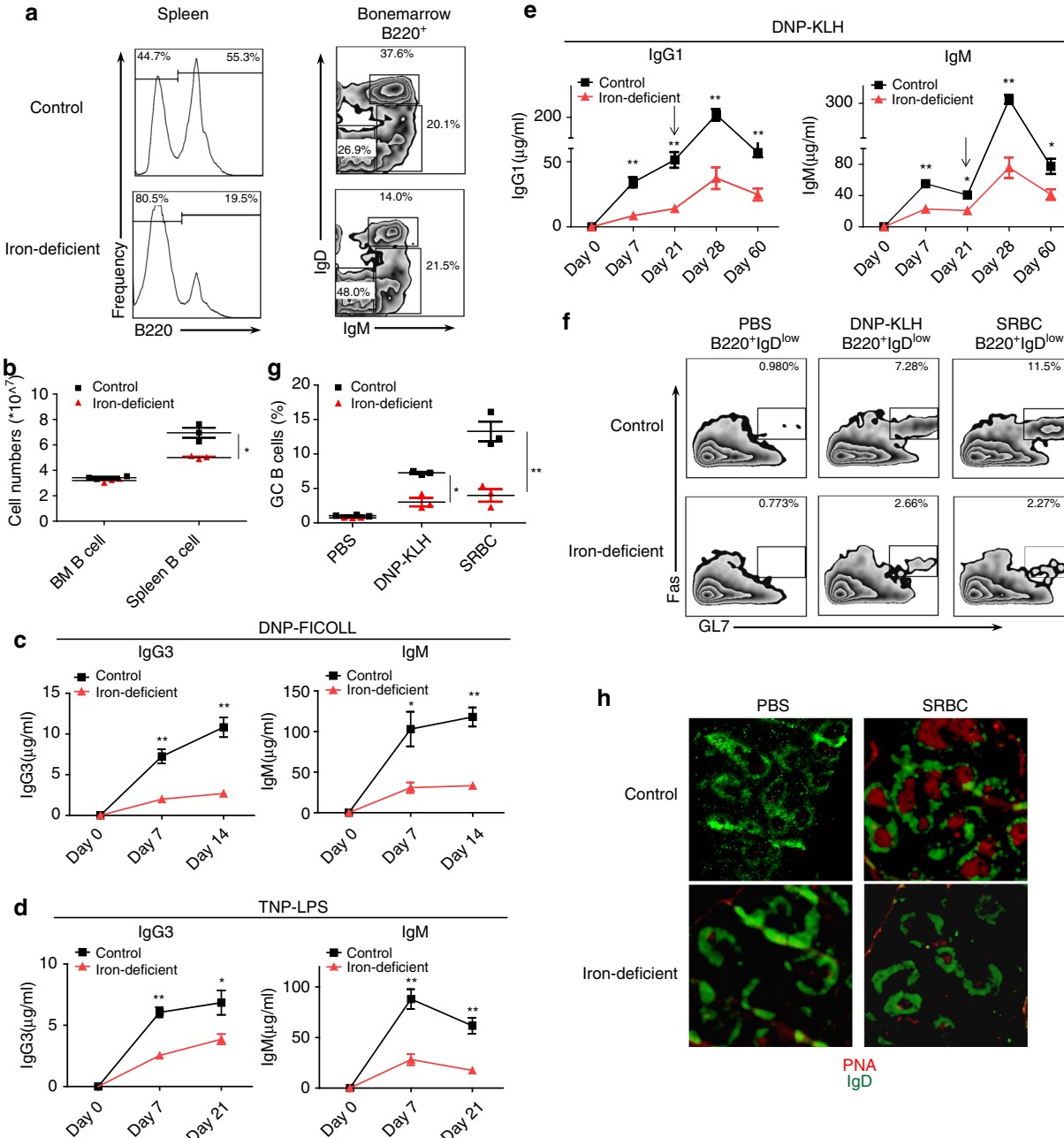

**Fig. 2** Iron deficiency results in impaired TD and TI immune responses in mice. Three-week-old wild-type C57BL/6J mice (male) were divided into two groups and fed either a control iron-adequate diet (45.4 mg/kg iron) or an iron-deficient diet (0.9 mg/kg iron) for 5 weeks. **a** Flow cytometry of splenocytes from control and iron-deficient mice stained with anti-B220 to identify peripheral B cells. Bone marrow cells from control and iron-deficient mice were stained with anti-B220, anti-IgM, and anti-IgD antibodies to assess pre- and pro-B cells (IgM−IgD−), immature B cells (IgM+IgDint), and mature circulating B cells (IgM+IgDhi). **b** Statistics for the numbers of splenic B cells in the two groups (mean ± SEM of three mice per group). **c**–**e** Antigen-specific antibody responses of control and iron-deficient mice immunized with DNP-KLH (TD), DNP-Ficoll (TI-2), or TNP-LPS (TI-1). Serum was collected preimmunization and at 7 days and 21 days after immunization. In the case of DNP-KLH, the mice were rechallenged at day 21 as indicated by the arrow, and serum was collected at day 28 and to a long-term at day 60 (the data represent the mean ± SEM of four mice per group). **f** Control and iron-deficient mice were immunized with SRBCs, DNP-KLH or PBS as a control, and at day 10, splenocytes were stained with anti-B220, anti-IgD, anti-GL7, and anti-CD95 (Fas) antibodies to assess the GCB cells (B220+IgDlowGL7+Fas+). **g** Statistics for the proportions of GCB cells in mice immunized with DNP-KLH or SRBCs (mean ± SEM of three mice per group). **h** Frozen sections of spleens from mice immunized with SRBCs were stained with anti-IgD antibody and PNA to identify germinal center reactions. The data were representative of at least two independent experiments. *P < 0.05, **P < 0.01, Student's t-test

CD45.1+ wild-type and CD45.2+ wild-type bone marrow (1:1), no difference was observed between CD45.1+ wild-type and CD45.2+ wild-type B-cell populations. However, mice that received CD45.1+ wild-type and CD45.2+ Steap3-KO bone marrow (1:1) had more CD45.1+ mature B cells than CD45.2+

Steap3-KO mature B cells, indicating that reduced number of mature B cells caused by *Steap3* depletion were intrinsic to B cells (Fig. 3e, f). Notably, neither type of bone marrow chimeric mouse displayed obvious differences in the percentages of MZB and FOB cells among CD45.1+ and CD45.2+ mature B cells

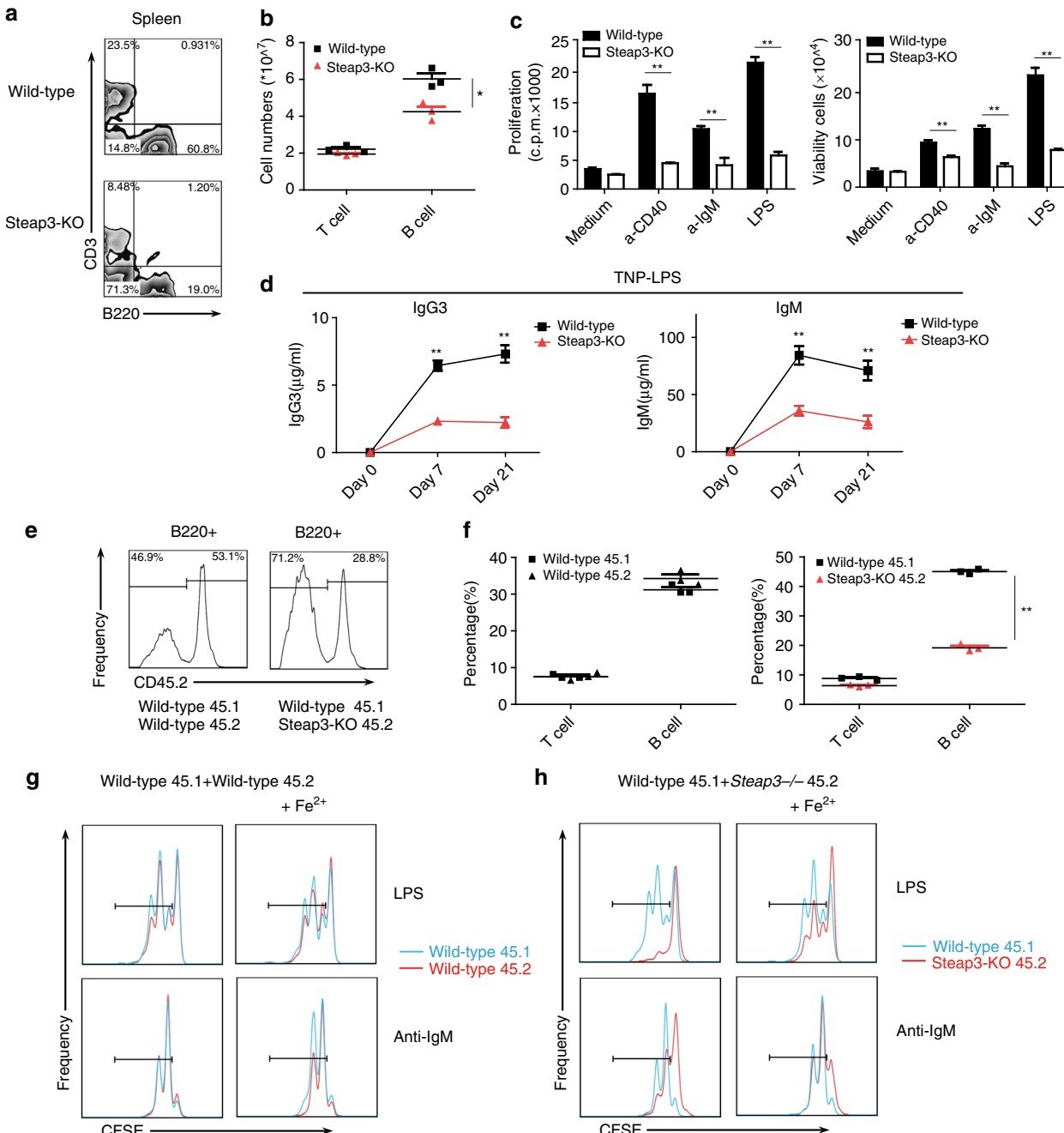

**Fig. 3** Defective mature B-cell population and proliferation in Steap3-KO mice. **a** Flow cytometry of splenocytes from wild-type and Steap3-KO mice was used to identify CD3 + T cells and B220 + B cells. **b** Statistics for the numbers of splenic B and T cells in the two groups (mean ± SEM of three mice per group). **c** Wild-type or Steap3-KO splenic B cells were stimulated with anti-CD40 (1 μg/ml), anti-IgM (10 μg/ml), or LPS (2 μg/ml), and proliferation was assessed by [³H] thymidine incorporation (**b**, top). Viable B cells were counted 48 h after stimulation (**b**, top). **d** ELISA to check antigen-specific antibody responses of wild-type and Steap3-KO mice 7 days and 21 days after immunization with TNP-LPS (the data represent the mean ± SEM of four mice per group). CD45.1 + wild-type bone marrow cells were mixed 1:1 with CD45.2 + wild-type or Steap3-KO bone marrow cells and cotransferred into irradiated wild-type C57BL/6J recipient mice. Six weeks later, **e** flow cytometry of splenocytes from recipient mice was conducted to compare the quantity of CD45.1 + wild-type B cells with that of CD45.2 + wild-type or Steap3-KO B cells. The splenocytes were further stained with anti-CD21 and anti-CD23 antibodies to evaluate MZB cells and FOB cells. **f** Statistics obtained for CD45.1 + or CD45.2 + T cells and B cells in the spleens of recipient mice (mean ± SEM of three mice per group). **g, h** Splenic B cells from recipient mice containing CD45.1 + and CD45.2 + B cells were labeled with CFSE, cultured and stimulated with anti-IgM (10 μg/ml) or LPS (2 μg/ml). The right panel shows the results after additional Fe2 + (ferrous iron) was added to the medium. B-cell proliferation was assessed by CFSE dilution, and the portion of cells that underwent at least one cellular division are outlined. The data were representative of three mice or two independent experiments. *$P < 0.05$, **$P < 0.01$, Student's $t$-test

(Supplementary Fig. 4e). Thus, *Steap3* depletion leads to mature B cell defects only with regards to total quantity, not with regards to development or maturation.

Then splenic B cells of bone marrow-transplanted recipient mice, including CD45.1$^+$ B cells (wild-type) and CD45.2$^+$ B cells (wild-type or Steap3-KO). These cells were labeled with CFSE and analyzed for cell proliferation in response to BCR or TLR stimulation (Supplementary Fig. 2d). As expected, the proliferation of wild-type CD45.1$^+$ B cells was similar to wild-type CD45.2$^+$ B cells (Fig. 3g); in contrast, very few Steap3-KO CD45.2$^+$ B cells underwent cell division in response to anti-IgM or LPS (Fig. 3h). Notably, wild-type CD45.1$^+$ B cells and Steap3-KO CD45.2$^+$ B cells were isolated from the same spleens, cultured in the same medium and exposed to the same stimuli. Thus, these data indicate that *Steap3* deletion resulted in B-cell proliferation defects.

As *Steap3* deletion abolished the ability of B cells to absorb and utilize extracellular Fe$^{3+}$ (to Fe$^{2+}$), we investigated whether iron deficiency was the cause for proliferation defects of Steap3-KO B cells. Additional Fe$^{2+}$ was supplemented in the B-cell culture medium before stimulation of cells with anti-IgM or LPS. Proliferation of wild-type CD45.1$^+$ and wild-type CD45.2$^+$ B cells was not markedly enhanced after Fe$^{2+}$ replenishment (Fig. 3g); however, Fe$^{2+}$ replenishment was able to mostly rescue the B-cell proliferation defects in Steap3-KO CD45.2$^+$ B cells in response to either BCR or TLR stimulation (Fig. 3h). These data indicate that the proliferation defects in Steap3-KO B cells were primarily caused by impaired iron uptake and iron deficiency in B cells.

**Iron is required for B-cell proliferation and plasma cell differentiation in vitro.** To further confirm the requirement of iron for B-cell proliferation, we used deferoxamine (DFO, a widely used iron chelator) to create an iron-deficient environment for cell culture in vitro. First, we measured the proliferative responses of B cells to CD40, BCR or TLR stimulation. Primary B cells cultured in the presence of DFO (iron-deficient B cells) proliferated poorly in response to anti-IgM or LPS stimulation compared with control cells, as assessed by tritiated thymidine incorporation. However, the addition of ferric ammonium citrate (FAC, Fe$^{3+}$) to replenish the iron fully restored B-cell proliferation (Fig. 4a and Supplementary Fig. 5a). Accordingly, supplementation with FAC could reverse the reduction in viability of iron-deficient B cells in response to anti-IgM or LPS stimulation (Fig. 4b).

We hypothesized that the inability of iron-deficient B cells to proliferate in response to BCR or TLR stimulation could be due to failure of the cells to survive or to enter the cell cycle. To test this hypothesis, B cells were labeled with CFSE and assessed for CFSE dilution in response to anti-IgM and LPS exposure (Supplementary Fig. 2e). As expected, most anti-IgM or LPS-stimulated control B cells underwent at least one cycle of cellular division, whereas iron-deficient B cells did not divide in response to anti-IgM or LPS stimulation, as revealed by the absence of CFSE dilution (Fig. 4c).

BAFF plays a crucial role in the survival and maintenance of mature B cells[18]. In the presence of BAFF, DFO did not appreciably alter the viability of B cells after 3 days. Anti-CD40 stimulation also promotes B-cell survival and mildly promotes cell division[19]; the percentage of live B cells (7-amino-actinomycin D negative, 7AAD$^−$) was similar between iron-deficient B cells and control B cells, but mild promotion of cell division was not observed in iron-deficient B cells. These data suggest that iron is probably not required for B-cell survival (Fig. 4d). B cells differentiate into antibody-secreting plasma cells after rounds of proliferation in response to antigenic challenge[20]; therefore, we assessed whether iron was required for B-cell differentiation into plasma cells (Supplementary Fig. 2f). As

expected, upon LPS stimulation, iron-deficient B cells could barely develop into CD138$^+$ plasma cells compared with control B cells; however, addition of FAC to iron-deficient B cells resulted in rescued plasma cell development that was almost equivalent to that of control B cells (Fig. 4e). These data demonstrate that iron is necessary for rapid B-cell population expansion and plasma cell formation in response to external stimuli; thus, iron deficiency in mice may lead to defects in germinal center formation and humoral immune responses.

To assess the situation in human B cells, we further check human B cells proliferation in iron-deficient conditions. Given that after we isolated human B cells from iron-deficient subjects, iron levels in B cells from iron-deficient patients would return to normal under normal culture conditions in vitro (rich iron ions in fetal bovine serum), we set up a new culture system to better mimic the proliferation of B cells in vivo in iron-deficient subjects and iron-normal subjects, in which we cultured B cells under 20% serum from the same individuals that B cells were isolated from. We selected two iron-normal subjects and two iron-deficient subjects (Fig. 4g). For B cells from iron-deficient subjects cultured in iron-deficient serum (cultured in 20% autologous serum conditions), significantly fewer cells undergo division and proliferation under LPS-stimulated conditions compared with B cells from iron-normal subjects (Fig. 4f, h). What's more, for B cells from iron-deficient subjects, FAC supplementation could dramatically increase the proliferation of human B cells in response to TLR stimulation; and for B cells from iron-normal subjects, B-cell proliferation was severely impaired after adding iron-chelating DFO in culture condition (Fig. 4i). These results further indicated a critical role of iron also in human B-cell proliferation defects.

**Iron is not required for early BCR signaling.** To further investigate the requirement of iron for B-cell activation and proliferation in response to BCR stimulation, we evaluated whether the impaired B-cell proliferation caused by iron deficiency was due to a defect in early BCR signaling events. As many BCR-initiated signaling events depend on calcium influx[21], we first measured intracellular calcium mobilization in response to BCR stimulation. Results showed no apparent proximal calcium mobilization defect in iron-deficient B cells (Fig. 5a). We also evaluated the downstream early phosphorylation of ERK1/2 after BCR engagement, and the results showed that early B-cell activation signals in response to BCR stimulation seemed to be intact in iron-deficient B cells (Fig. 5b).

We further designed a time-course iron deprivation assay and, correspondingly, a time-course iron supplementation assay to determine which period of BCR or TLR signaling required iron for B-cell proliferation. In the time-course iron deprivation assay, B-cell proliferation in response to BCR or TLR stimulation was assessed at 72 h. During this process, DFO was added at different time points to chelate iron in the normal medium. The results showed that iron deprivation via addition of DFO within 24 h post stimulation completely suppressed B cell proliferation; however, iron deprivation after 48 h post stimulation did not affect B-cell proliferation (Fig. 5c). Interestingly, in the time-course iron supplementation assay, replenishment of DFO-treated B cells (iron-deficient B cells) with FAC (Fe$^{3+}$) within 24 h post stimulation could fully restore B-cell proliferation, while addition of FAC (Fe$^{3+}$) to iron-deficient cultures at 48 h post stimulation could only partially attenuate the defects in iron-deficient B-cell proliferation (Fig. 5d). These results indicate that iron is required only for the middle and late stages (between 24 h and 48 h post stimulation) of BCR- or TLR-induced B-cell activation and proliferation but not for the early stage of signaling.

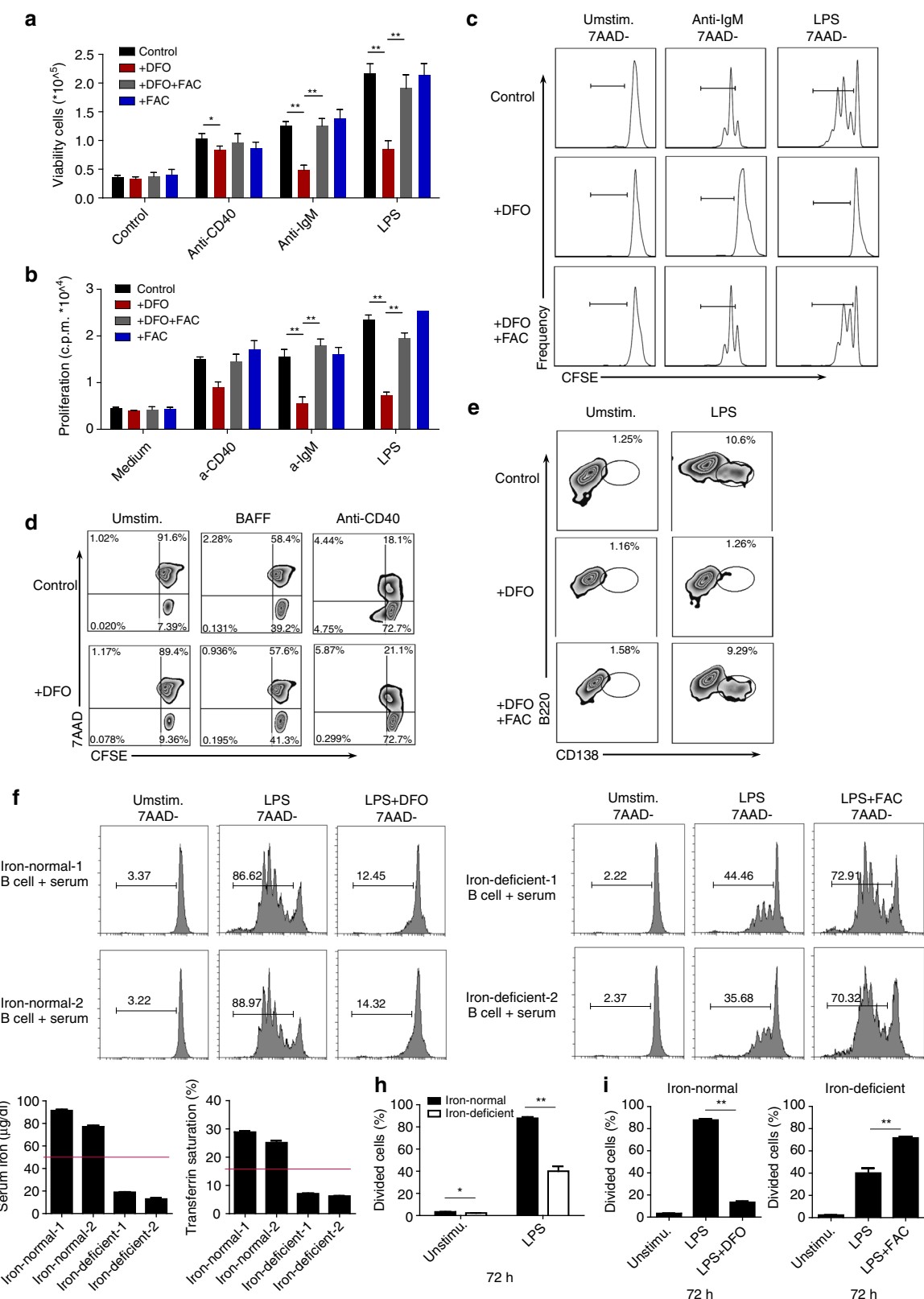

**Iron is responsible for S phase entry and cyclin E1 induction during B-cell proliferation**. To understand the mechanisms by which iron enables B-cell proliferation, we further analyzed global gene expression differences between iron-deficient B cells and control B cells by microarray analysis. Forty-eight hours after BCR or TLR activation, pathway analysis revealed that a series of cell cycle-related genes were downregulated in iron-deficient B

cells compared with control B cells (Fig. 6a). Thus, we conducted a 5-bromo-2-deoxyuridine (BrdU) incorporation assay to assess cell cycle progression in response to anti-IgM or LPS and found that iron-deficient B cells exhibited G1/S arrest (Fig. 6b).

Cyclins play crucial roles in the regulation of cell cycle entry and cell cycle progression, and two cycle-related genes, cyclin D and cyclin E, are critical for S phase entry of activated B cells[22].

**Fig. 4** Iron is required for rapid B-cell proliferation and plasma cell formation in vitro. **a** Proliferation of splenic B cells cultured in normal medium or in the presence of DFO (20 µM), DFO accompanied by FAC (100 µM), or FAC alone (100 µM) and unstimulated or stimulated with anti-CD40 (1 µg/ml), anti-IgM (10 µg/ml) or LPS (2 µg/ml) for 72 h. B-cell proliferation was assessed by [$^3$H] thymidine incorporation. **b** The number of viable B cells in **a** was counted. **c** Cell division (CFSE dilution) of CFSE-labeled B cells unstimulated or stimulated with anti-IgM (10 µg/ml) or LPS (2 µg/ml) under different iron conditions. Cells that underwent at least one cellular division are outlined. **d** Cell survival (7AAD-) and cell division of CFSE-labeled splenic B cells unstimulated or stimulated with anti-CD40 (1 µg/ml) or BAFF (100 ng/ml) in the presence or absence of DFO (20 µM). **e** Splenic B cells were purified from wild-type C57BL/6 mice, stimulated for 5 days with LPS under different iron conditions, and stained with anti-B220 and anti-CD138 (a plasma cell marker). The outlined areas indicate plasma cells (B220$^{low}$CD138$^+$). **f** Human B cells were isolated from peripheral blood of iron-deficient subjects and iron-normal subjects, cultured in 20% autologous serum conditions and cell proliferation in response to TLR stimulation were assessed. **g** Serum iron level and TS examination data of the two iron-deficient subjects and two iron-normal subjects. Red line indicates minimum normal value of serum iron and TS in human. **h, i** Statistics for divided B cells in **f**. The data were representative of two independent experiments. *$P < 0.05$, **$P < 0.01$, Student's $t$-test

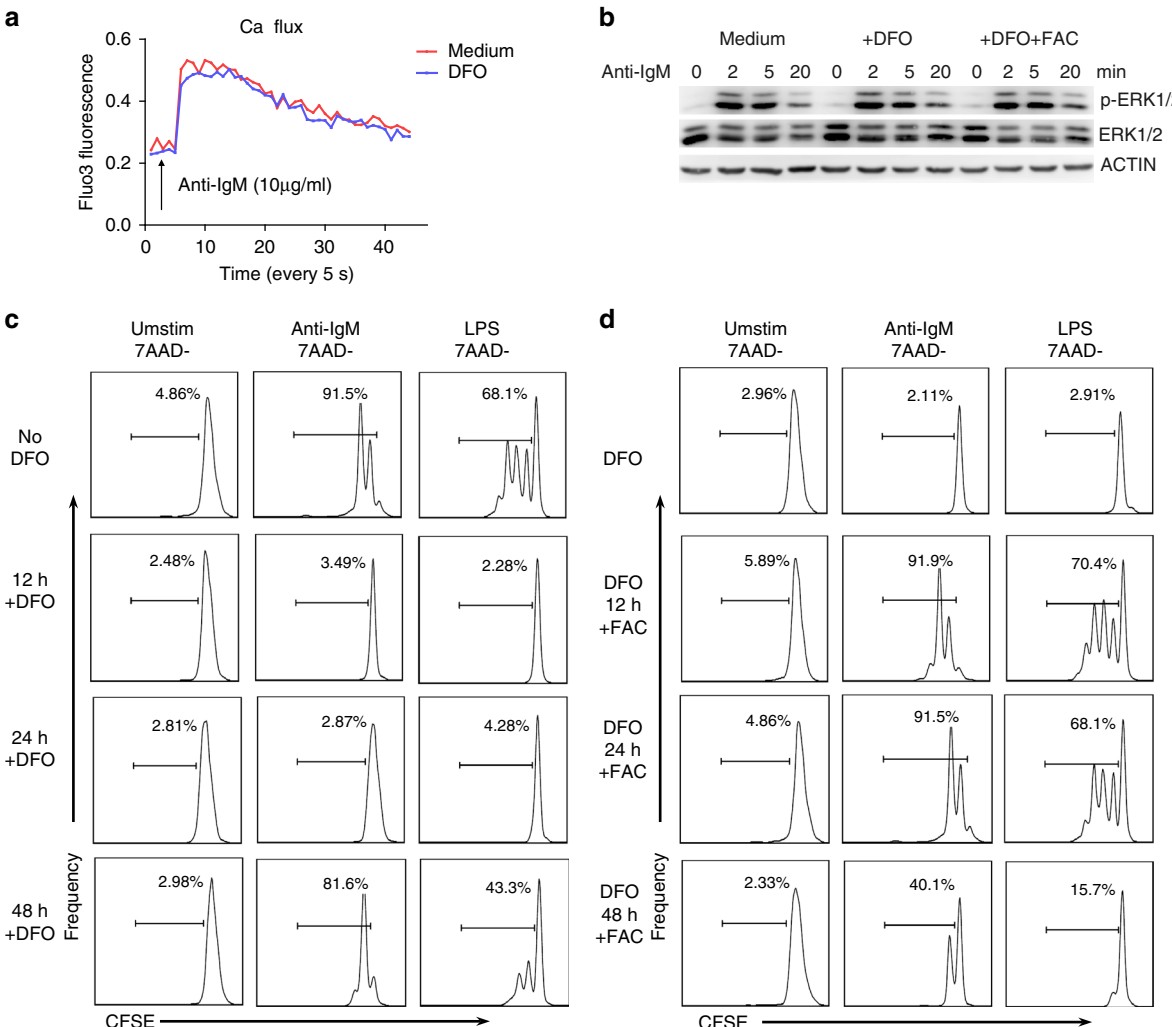

**Fig. 5** Iron is essential only for the middle and late stages of BCR- and TLR-induced B-cell proliferation. **a** Intracellular calcium flux in control and iron-deficient (pretreated with 20 µM DFO for 2 h) resting B cells purified from wild-type C57BL/6 splenocytes in response to BCR stimulation. **b** Control and iron-deficient (pretreated with 20 µM DFO for 2 h) B cells were stimulated with anti-IgM (10 µg/ml) for 0–20 min. At the indicated time points, the cells were immediately collected, washed, and lysed. Subsequently, the phosphorylation of ERK1/2 was measured by immunoblotting. **c** CFSE-labeled splenic B cells from wild-type mice were unstimulated or stimulated with anti-IgM and LPS, and cell proliferation was assessed by CFSE dilution at 72 h. During this process, DFO (20 µM) was added to the culture medium at different time points to deplete iron. **d** CFSE-labeled iron-deficient B cells (in the presence of 20 µM DFO) were stimulated as in **c**, and cell proliferation was assessed at 72 h. During this process, FAC was added to the culture medium at different time points to restore the iron supply. The data are representative of three independent experiments

Detailed analysis confirmed that cyclin D2 and cyclin E1 were the major isoforms associated with B-cell proliferation (Supplementary Fig. 6a). *Cyclin E1* expression was markedly induced 24 h and 48 h after BCR or TLR stimulation in control B cells but was severely impaired in iron-deficient B cells. In contrast, induction of *cyclin D2* occurred ~4 h post stimulation and remained unaffected 12, 24, and 48 h post stimulation in iron-deficient B cells (Fig. 6c, d and Supplementary Fig. 6b, c). Additionally, Bcl-x$_L$ (*Bcl2l1*) induction was intact in iron-deficient B cells, consistent with the results that iron was not

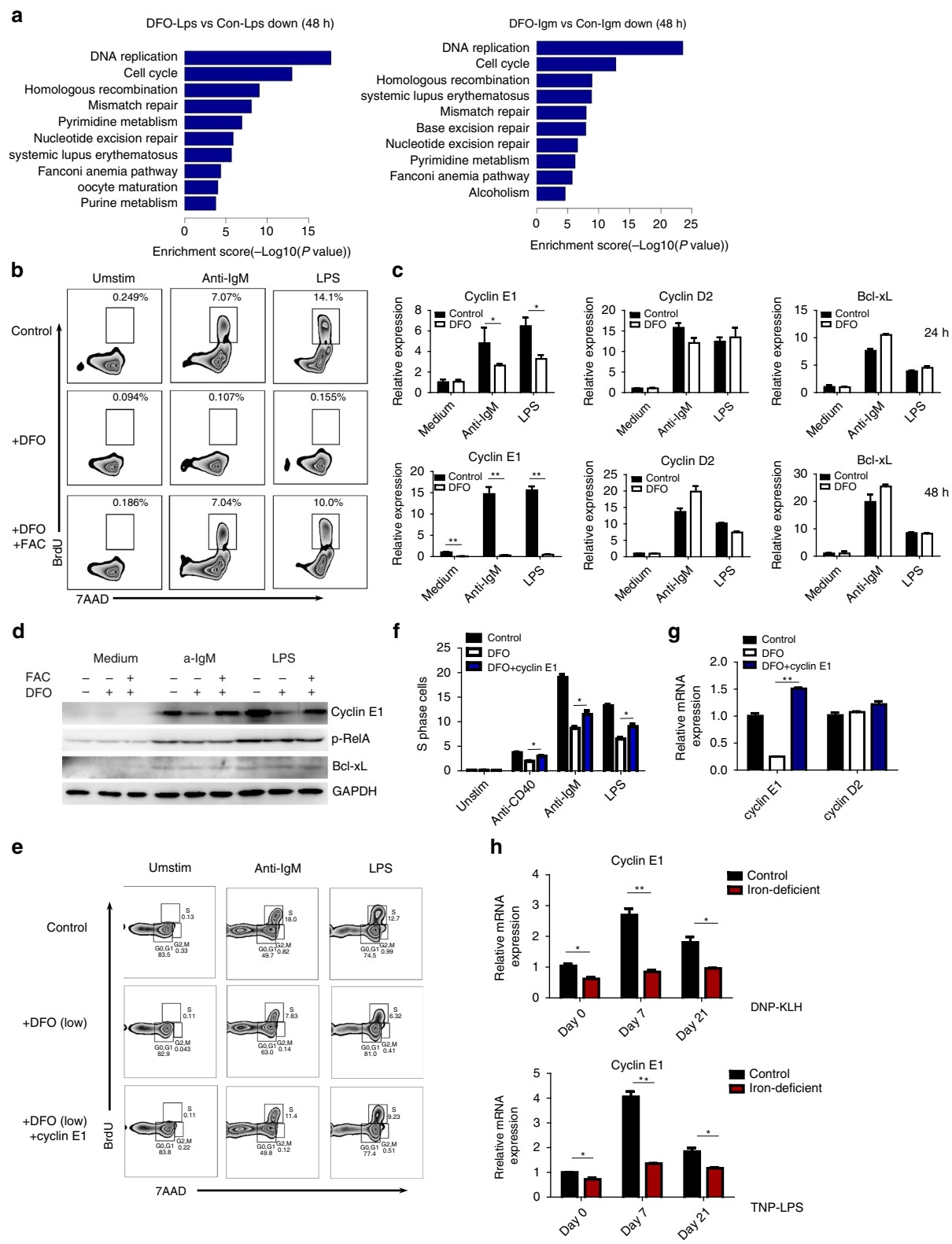

required for B-cell survival (Fig. 6c, d). These results also suggest that iron is not required for gene induction at the early stage of BCR or TLR signaling. Moreover, FAC supplementation 20 h post stimulation completely restored cyclin E1 induction in iron-deficient B cells, further verifying that the requirement of iron for gene regulation mostly applies to the middle and late stages of BCR- and TLR-induced B-cell proliferation (Supplementary Fig. 6d).

We knocked down cyclin E1 through short hairpin RNA (shRNA) lentivirus transduction into primary B cells to assess the influence of cyclin E1 on B cell proliferation (Supplementary Table 1). Although the transduction efficiency was limited, interference with cyclin E1 expression in B cells could inhibit their proliferation in response to BCR or TLR stimulation (Supplementary Fig. 7d, e and f). Furthermore, overexpression of cyclin E1 could significantly attenuate the S phase entry defects in

**Fig. 6** Iron is responsible for S Phase entry and cyclin E1 induction during B-cell proliferation. **a** Microarray analysis of global gene expression in control and iron-deficient B cells that were unstimulated or stimulated with anti-IgM (10 μg/ml) or LPS (2 μg/ml) for 48 h. Pathway analysis was conducted to map genes that were associated with Kyoto Encyclopedia of Genes and Genomes (KEGG) pathways. **b** B cells cultured and stimulated as in **a** were cocultured with 10 μM BrdU for the last 45 min, stained with anti-BrdU and 7AAD, and analyzed by flow cytometry. The outlined areas represent the percentage of cells in S phase (with BrdU incorporation during DNA replication). **c** The mRNA levels of cyclin D2, cyclin E1, and Bcl-xL were measured by qRT-PCR in control and iron-deficient B cells that were cultured and stimulated as described in **a**. **d** Immunoblot analysis of cyclin E1, Bcl-xL, and phos-RelA in splenic B cells cultured in normal medium or in the presence of DFO (20 μM) or DFO with FAC (50 μM) under the indicated conditions for 48 h. **e** Splenic B cells from wild-type C57BL/6 mice in low-DFO conditions (10 μM) were infected with cyclin E1 overexpression lentivirus, and BrdU incorporation was analyzed by flow cytometry following anti-IgM (10 μg/ml) or LPS (2 μg/ml) stimulation for 48 h. **f** Statistics for the percentage of B cells proceeding through DNA replication and entering S phase. **g** Cyclin E1 and cyclin D2 mRNA were checked by qRT-PCR in control and cyclin E1-overexpression B cells. **h** qRT-PCR of cyclin E1 mRNA expression in splenic B cells isolated from control and iron-deficient mice immunized with DNP-KLH or TNP-LPS. The data are representative of two independent experiments. All graphs with error bars indicate the mean ± SEM, *$P < 0.05$, **$P < 0.01$, Student's $t$-test

iron-deficient B cells (Fig. 6e–g). To further confirm the regulation of iron on cyclin E1 expression, we isolated splenic B cells from control and iron-deficient mice immunized with T-cell-dependent antigen DNP-KLH and T-cell-independent TI-1 antigen TNP-LPS, and checked the cyclin E1 expression by quantitative PCR (qPCR). Data indicated that cyclin E1 induction was significantly decreased in iron-deficient mice, which further support results of our conclusion based on in vitro experiments (Fig. 6h). Altogether, these data demonstrate that cyclin E1, which is induced during the middle and later stages of BCR and TLR signaling, may be a critical iron-dependent target gene responsible for proliferation defects in iron-deficient B cells.

**H3K9 demethylation of cyclin E1 is regulated in an iron-dependent manner**. As eukaryotic cells primarily absorb and utilize iron after it is reduced from $Fe^{3+}$ to $Fe^{2+}$, we searched for biological processes that might require iron, particularly $Fe^{2+}$. Members of a class of JmjC domain-containing proteins have been reported to be histone demethylases that catalyze the demethylation of methylated lysines on histone tails through oxidative reactions. Notably, this reaction is based on an $Fe^{2+}$-dependent catalytic center and involves 2-OG as a cofactor[23]. Given that $Fe2+$ is essential for JmjC demethylase to function, we hypothesized that the regulation of cyclin E1 induction and B-cell proliferation by iron might occur through a JmjC demethylase. To test this hypothesis, we treated B cells with IOX-1, an inhibitor of 2-OG, which is another essential member of the active JmjC domain-containing demethylase family[24]. We found that 2-OG inhibition resulted in impaired B-cell proliferation that was almost equivalent to that resulting from iron deprivation (Fig. 7a). Additionally, gene expression profiles were similar in 2-OG-inhibited B cells and iron-deficient B cells, and induction of cyclin D2 and Bcl-x$_L$ in response to BCR or TLR activation remained intact, whereas cyclin E1 induction was severely inhibited (Fig. 7b). These data suggest that JmjC demethylase-mediated histone demethylation processes might play roles in B-cell proliferation.

Referring to the microarray data and conducting quantitative real-time PCR (qRT-PCR) to confirm, we validated that 5 of the 30 demethylases were significantly upregulated after BCR or TLR stimulation (Supplementary Fig. 8a). To determine which demethylases might be involved in B-cell proliferation, we constructed shRNAs to knockdown these five demethylases. Among the five demethylases, knocking down KDM2B, KDM3B, or KDM4C inhibited B-cell proliferation in response to BCR or TLR stimulation to some extent, whereas knockdown of KDM6B or PHF2 seemed to have little effect on B-cell proliferation (Supplementary Fig. 8b, c). Given that these three demethylases have been reported to mediate H3K9 demethylation[23,25], chromatin immunoprecipitation-sequencing (ChIP-seq) analysis

of control and iron-deficient B cells subjected to BCR activation was performed. To correlate chromatin bindings with direct gene regulation, we integrated the iron-dependent transcriptome with epigenome and noticed that a total of 127 genes might be directly regulated by iron-dependent demethylases, as they showed a reduction of expression levels and increased H3K9me2 modification in the absence of iron (Fig. 7c). The results revealed that H3K9me levels near the transcription start site (TSS) of cyclin E1 were significantly decreased in B cells after BCR stimulation; this finding was consistent with the induction of cyclin E1 expression because H3K9me is generally associated with gene repression. However, in accordance with the impaired induction of cyclin E1, this demethylation process did not occur in iron-deficient B cells; we speculate that this lack of demethylation is probably because the enzymatic activity of certain JmjC demethylases was inhibited in iron-deficient B cells (Fig. 7d). Interestingly, H3K9me levels were relatively low in chromatin regions near the TSS of cyclin D2, which might explain the phenomenon that cyclin D2 was not silenced and could be induced normally in iron-deficient B cells (Fig. 7d).

We next used ChIP-quantitative PCR to further validate these histone methylation changes. Within ~1 kilobase upstream or downstream of the TSS of cyclin E1, BCR or TLR stimulation greatly reduced the levels of H3K9 tri- and di-methylation (H3K9me3/2) in control B cells, whereas in iron-deficient B cells, H3K9me demethylation of cyclin E1 was severely inhibited (Fig. 7e). Moreover, 48 h after BCR or TLR stimulation, H3K9me3/2 demethylation of cyclin E1 was significantly inhibited in IOX1-treated B cells, similar to the situation in iron-deficient B cells, which also exhibited impaired induction of cyclin E1 after 2-OG inhibition and cell activation (Fig. 7f).

To further confirm the functions of the three JmjC demethylases, KDM2B, KDM3B, and KDM4C, in B-cell proliferation, we knocked down all three together (triple-KD) in splenic B cells. Triple-KD of the three JmjC demethylases resulted in significantly inhibited cyclin E1, but not cyclin D2, induction in response to BCR or TLR stimulation (Fig. 8a). Consequently, the proliferation of B cells was also markedly inhibited (Fig. 8b, c). We further observed increased H3K9 methylation at the promoter region of cyclin E1 in triple-KD B cells similar as the situation in iron-deficient B cells (Fig. 8d). These results indicated that iron is required for cyclin E1 induction primarily because it alters H3K9me status and that the three demethylases KDM2B, KDM3B, and KDM4C might be involved in this process.

## Discussion

In the present study, the trace element iron was found to be required for B-cell proliferation and for production of antibodies to antigens. Thus, we have revealed a critical role for iron in humoral immunity.

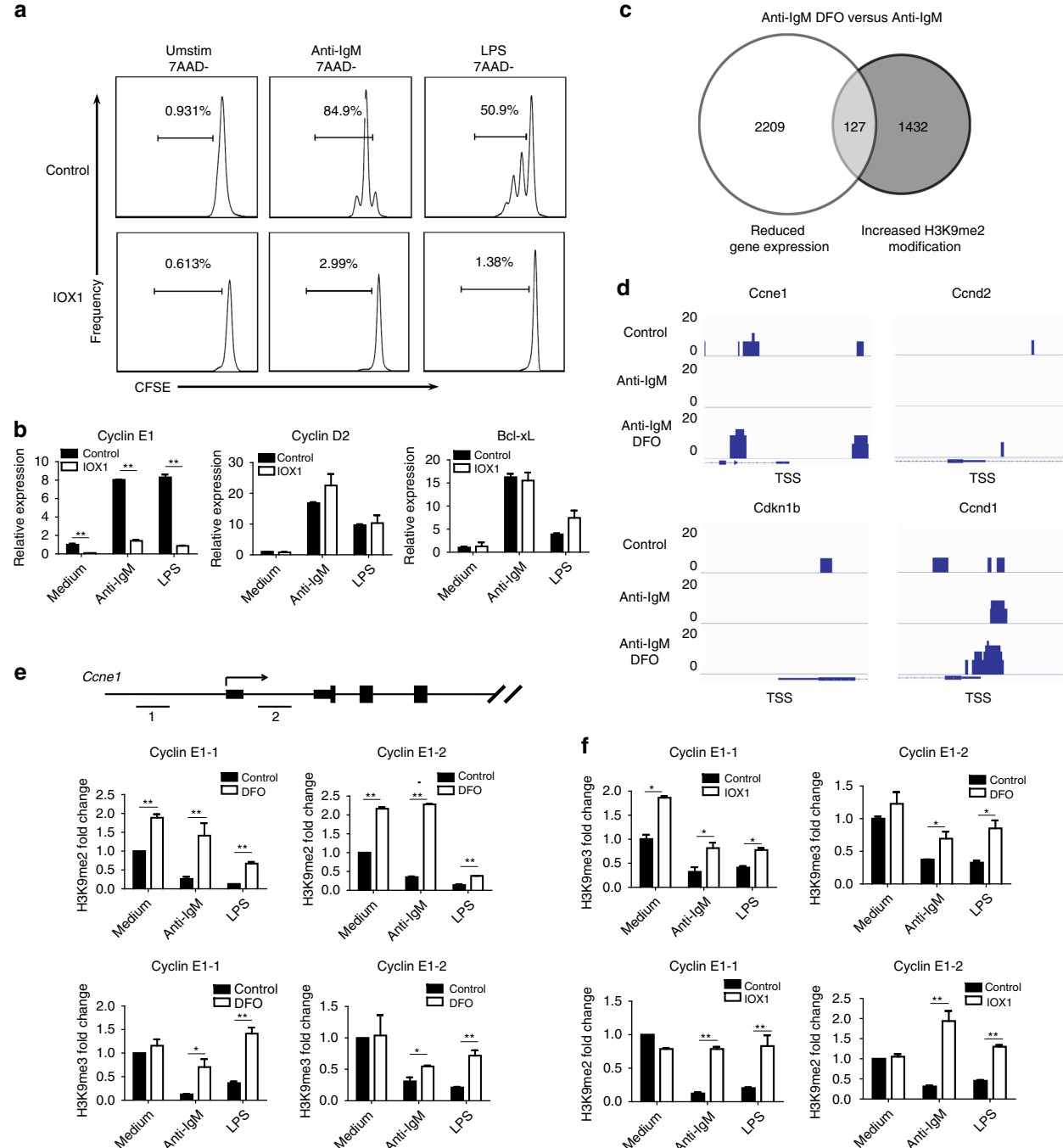

**Fig. 7** Iron deprivation impairs Cyclin E1 induction through inhibiting H3K9me3/2 demethylation. **a** Flow cytometry of CFSE-labeled B-cell proliferation following stimulation with anti-IgM (10 μg/ml) or LPS (2 μg/ml) for 72 h in the presence of the 2-OG inhibitor IOX1 (100 μM) or dimethyl sulfoxide (DMSO) as a control. **b** Results of qRT-PCR to analyze cyclin E1, cyclin D2, and Bcl-xL induction 48 h after anti-IgM or LPS stimulation in control and 2-OG-inhibited B cells. **c** Venn diagram indicating the overlap of genes with reduced expression and increased H3K9me2 modifications in iron-deficient B cells. **d** ChIP-seq analysis of H3K9me2 modifications in control and iron-deficient B cells post BCR stimulation. The results for four representative cell cycle-related genes are shown in the graph. **e** Results of ChIP-qRT-PCR to confirm changes in H3K9me3/H3K9me2 modification near the TSS of cyclin E1 in control and iron-deficient B cells following BCR or TLR stimulation for 48 h. The numbers and horizontal lines show the locations of the ChIP-qRT-PCR primers. **f** ChIP-qRT-PCR to analyze changes in H3K9me3/2 near the cyclin E1 TSS in control and 2-OG-inhibited B cells 48 h post stimulation. All graphs with error bars indicate the mean ± SEM, and *$P < 0.05$, **$P < 0.01$, Student's $t$-test

In clinical vaccination, low levels of antibody production are regarded as vaccine failure, which indicates that a group of individuals has a marginal or insufficient antibody response after immunization with a vaccine. The lack of efficacy of the measles and varicella vaccine is concerning, as there have been reports of outbreaks despite high vaccination coverage[26–28]. Hepatitis B vaccines are unable to elicit adequate antibody production in ~5% of vaccinated individuals[29,30]. In recent decades, countless efforts have been undertaken to enhance antibody responses to vaccines to reduce the risk of infection and disease, primarily involving developing new generations of vaccines or improving immunization strategies. Our data highlight that serum iron correlates

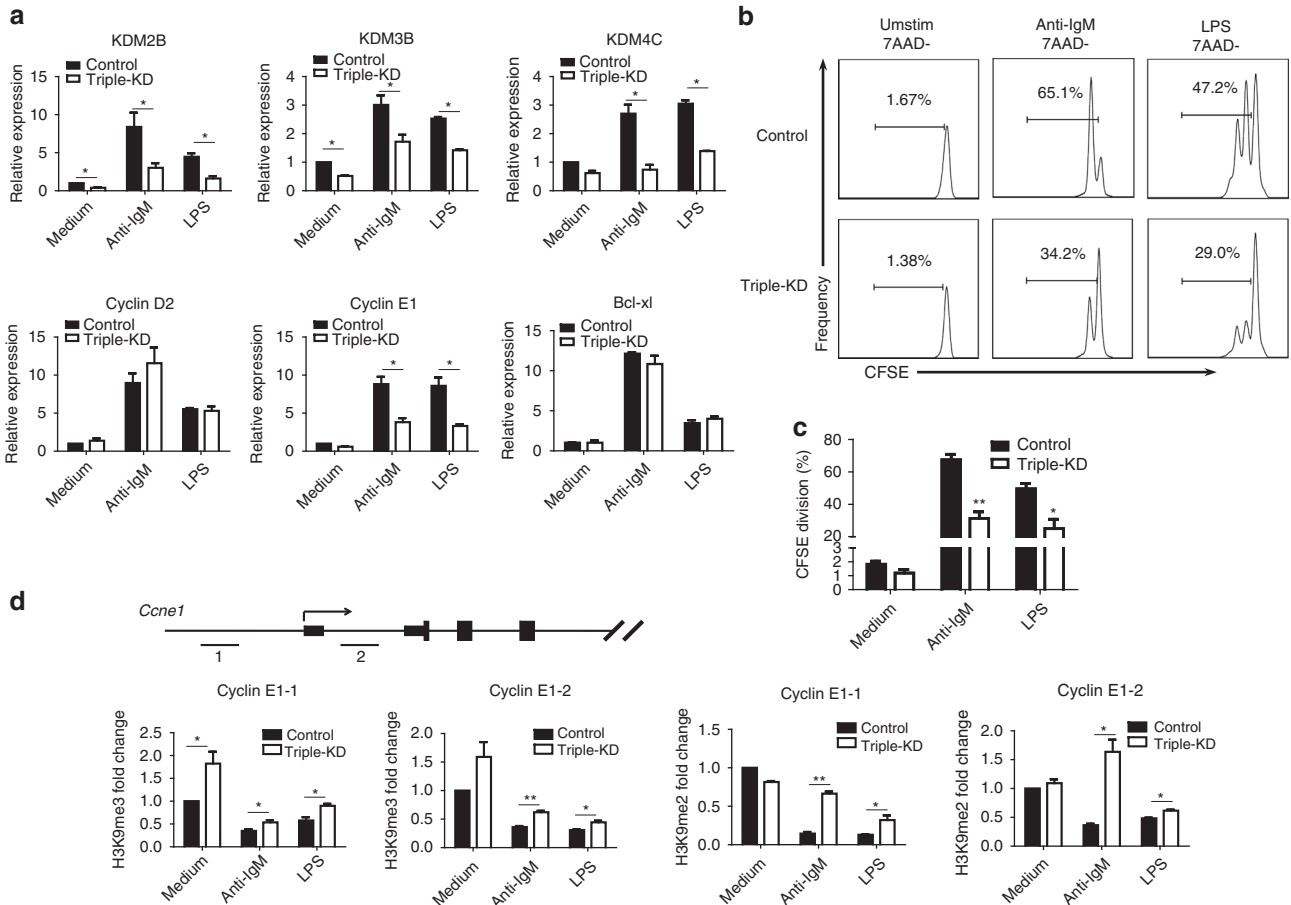

**Fig. 8** Triple-KD of three JmjC-demethylases reduces H3K9me3/2 demethylation and inhibits B-cell proliferation. shRNA lentiviruses targeting KDM2B, KDM3B, and KDM4C were mixed and used to infect splenic B cells. **a** The expression of cyclin E1, cyclin D2, Bcl-xL, and the three demethylases was analyzed by qRT-PCR 48 h after anti-IgM and LPS stimulation. **b** CFSE-labeled splenic B cells were infected with the shRNA lentivirus mixture, and cell proliferation was analyzed by flow cytometry at 72 h after anti-IgM and LPS stimulation. **c** Statistics for the percentage of B cells that underwent division following infection with the shRNA lentivirus mixture or control. **d** Changes in H3K9me2 at the cyclin E1 promoter were analyzed by ChIP-qPCR. All graphs with error bars indicate the mean ± SEM, *$P < 0.05$, **$P < 0.01$, Student's $t$-test

closely with measles vaccine responses in humans and that iron is necessary for B-cell proliferation, differentiation into plasma cells and consequent antibody production in a mouse model. Thus, serum iron examination and iron replenishment in iron-deficient individuals before injection of the vaccine might be necessary to improve vaccine effectiveness. These findings might also explain, to some extent, why ensuring adequate supply of the trace element iron could help provide protection against pathogen infection, likely by enabling the production of sufficient antigen-specific antibodies.

In a study using large cohorts of normal individuals in China, Xu and colleagues[31] found high prevalence of STEAP3 mutation (5.3% in 2338 individuals). Sixteen different loss-of-function STEAP3 mutations were identified, which resulted in severely or moderately impaired ferrireductase activity[31]. However, the deleterious effect in humans of these common abnormalities remains to be confirmed. Considering our study showed that Steap3-KO B cells from mice model exhibit severe defects in B-cell proliferation and immune function, whether these people with STEAP3 mutations (especially those with loss-of-function mutations) present immune B cell defects seems to be an interesting question worth further exploring.

Certainly, iron deficiency may impair other cell functions, including T-cell functions, that are indispensable for antibody

production, particularly with regard to TD antigens, since histone demethylation plays a critical role in T-cell differentiation and function[32]. We also measured proliferative responses of T cells to TCR stimulation in different concentrations of DFO, and results revealed that the proliferative responses of B cells to TLR stimulation was most sensitive to iron deprivation, the second-most sensitive was B cells to BCR stimulation, and proliferative responses of T cells to TCR stimulation was relatively less sensitive to iron deprivation, suggesting the different dependency of T-cell and B-cell function on iron (Supplementary Fig. 5b). Moreover, iron is also responsible for cyclin E1 induction of T cells in response to TCR stimulation, this may explain why iron can also to some extent inhibit T-cell proliferation and activation (Supplementary Fig. 5c). This finding suggests that iron deficiency in humans might result first in impairment of antibody responses to TI antigens, next in impairment of antibody responses to TD antigens, and last in impairment of T-cell immunity.

Given that iron participates in many oxidation-reduction processes, iron may enable DNA synthesis by regulating oxidation-reduction processes[33–35]. However in our study, data revealed that the two genes critical for S phase entry in B lymphocytes, cyclin D2, and cyclin E1, were significantly induced at 24 h and 28 h post stimulation, when the DNA synthesis was not

detected (Supplementary Fig. 7a, b). This finding indicates that induction of the critical cell cycle-related gene cyclin E1 occurs prior to DNA synthesis and S phase entry in activated B cells. In consideration of the cyclin E1 rescue experiment, overexpression of cyclin E1 in iron-deficient B cells could notably increase the proportion of B cell that entered into S phase. These data indicate that the iron-dependent induction of cyclin E1 plays an essential role during B-cell proliferation, although we cannot rule out the possibility that iron may function in DNA synthesis after cycle-related gene induction during B-cell proliferation.

In this study, we demonstrated the critical role of iron in B-cell proliferation, which is critical and essential for downstream events of B cells, including germinal center formation, antibody production, memory B-cell development. Generally, germinal center formation is essential for memory B-cell development[36,37]. We did not set-up a system to specifically check measles-specific memory B-cell development in iron-deficient mice. The observation that dramatically reduced germinal center formation, deficient primary, and recall humoral immunity after antigen immunization existed in iron-deficient mice, suggesting that iron-dependent H3K9 demethylation might be critical for memory B-cell development. In order to access whether iron was related to the generation of memory B cells or long-lived plasma cells, we need to set-up a new experimental system in which iron deficiency bypasses deficient B-cell proliferation.

Taken together, our findings provide new insights into the functions of the trace element iron in humoral immunity that will be beneficial for improving vaccine efficacy. Although iron deficiency may influence B-cell proliferation through multiple mechanisms, we demonstrate that iron-dependent H3K9 demethylation mediated by KDM2B, KDM3B, and KDM4C plays a critical role in cyclin E1 induction and B-cell proliferation.

## Methods

**Population study and MV response test**. The samples assessed in this study were selected from a previous seroprevalence survey of measles in Zhejiang Province, China. The survey consisted of a multistage design and was employed to obtain samples from five counties in Zhejiang Province after consideration of geographical and economic status. Measles in Zhejiang Province is at a low level, and these subjects also come from places where measles has no epidemic or very low levels of disease in recent years, so as to avoid interference with disease infection. Serum samples from individuals who were known to be affected by an acute infection were excluded. Measles vaccine is routine immunization of the population. In recent decades, Zhejiang Province routinely use two needles of measles vaccination for children aged 8 months and 18 months, with a dose of 0.5 ml. The source of the vaccine is China Biotechnology Co., Ltd. Serum samples from participants older than 10 years were selected for this study. Written informed consent was obtained from all participants or their guardians (for children younger than 18 years old) before enrollment in the study. The study was approved by the ethics committee of Zhejiang Provincial Center for Disease Control and Prevention, and complied with the principles expressed at the Declaration at Helsinki.

Serum samples were stored at $-20\,°C$ and transferred to ice before analysis. Serological tests were performed at the measles laboratory of the Zhejiang Provincial Center for Disease Control and Prevention, which met the accreditation criteria of the World Health Organization (WHO) National Measles Laboratories. A commercial enzyme-linked immunosorbent assay (ELISA) kit for measles (SERIO ELISA classic Measles Virus IgG, Institute Virion/Serion GmbH) was used to quantitatively and qualitatively measure the presence of specific IgG antibodies against measles virus in sera. Cutoff and final results were based on the qualitative criteria outlined by the manufacturer, and standard controls in duplicate and negative controls were used in every plate. Serum iron and unsaturated iron binding capacity (UIBC) were quantified using a colorimetric assay kit (Iron/UIBC, Pointe Scientific, Inc). TS (%) = serum iron/(serum iron + UIBC) × 100.

**Inductively coupled plasma-mass spectrometer (ICP-MS) analysis**. Serums were diluted 20-fold with diluents (0.1% triton X 100, 0.1% HNO3 in DI water), and subject to metal analysis with ICP-MS (Agilent 7900, Agilent Technologies, Santa Clara, CA).

**Mice**. Steap3-KO mice on a C57BL/6 background were obtained from Dr. Wang Fudi (Zhejiang University), and genotyping was performed as previously reported. Wild-type C57BL/6 mice were used to construct the mouse model of iron

deficiency, and isolated primary B cells were purchased from the Shanghai Laboratory Animal Center, Chinese Academy of Sciences, Shanghai. All animals were housed and bred under specific pathogen-free conditions. All animal experiments were performed in compliance with the NIH Guide for the Care and Use of Laboratory Animals (National Academies Press, 2011) and were approved by the Institutional Biomedical Research Ethics Committee of the Shanghai Institutes for Biological Sciences, Chinese Academy of Sciences.

**B-cell isolation and in vitro proliferation**. Splenic B cells were isolated with MACS beads by negative selection according to the manufacturer's protocol (STEMCELL Technologies). The purity was routinely >96% as assessed by staining with anti-B220. For the in vitro proliferation assay, splenic B cells were labeled with 1 mM CFSE (Sigma-Aldrich) for 10 min in phosphate-buffered saline (PBS) and washed twice with RPMI 1640 medium containing 10% fetal bovine serum (FBS) before plating. B cells at a density of $1 \times 10^6$ cells/ml were then cultured in normal culture medium and stimulated with F(ab')$_2$ goat anti-mouse IgM (10 μg/ml; Jackson ImmunoResearch), LPS from *Escherichia coli* O111:B4 (2 μg/ml; Sigma-Aldrich), recombinant BAFF (100 ng/ml; R&D Systems) and anti-CD40 (1 μg/ml; BD Biosciences). Normal culture medium for B cells was 1640 medium with 10% (vol/vol) FBS, 0.05 mM β-mercaptoethanol, 1 mM sodium pyruvate, 100 U/ml penicillin and 100 μg/ml streptomycin. CFSE dilution was analyzed by flow cytometry after 72 h of stimulation. In some experiments, 20 μM DFO or 100 μM FAC was applied to the cell culture medium. For the [$^3$H]thymidine incorporation assay, purified B cells were also plated at a density of $2 \times 10^5$ cells per well (200 μl in 96-well plates) with various levels of stimulation and cultured for 72 h with the addition of 0.4 mCi [$^3$H]thymidine for the final 12 h of culture.

**Flow cytometry**. Single-cell suspensions of spleen and bone marrow cells from femurs were incubated in Red Blood Cell Lysis Buffer (ACK lysis buffer to deplete red blood cells) for 5 min. Spleen or bone marrow cells were then stained in 100 ml of staining buffer (PBS + 0.5% bovine serum albumin (BSA)) containing various fluorescent antibodies. After incubation on ice for 30 min, the cells were collected using a FACSCalibur or FACSAria (BD Biosciences), and the data were analyzed using FlowJo software.

**Bone marrow transfer**. Bone marrow cells were flushed from the femur and tibia bones of adult Steap3-KO or control mice (8-weeks-old). Subsequently, erythrocytes were lysed by incubation in ACK lysis buffer for 5 min, and the number of cells was counted before the transfer. Bone marrow cells ($2.5 \times 10^6$) from control CD45.1 mice were mixed (1:1 ratio) with bone marrow cells from control CD45.2 mice or Steap3-KO mice and transferred into recipient wild-type C57BL/6 mice that had been irradiated with 8.5 Gy. After 6 weeks, the recipient mice were euthanized, and peripheral mature B cells were analyzed as previously described.

**RNA extraction and qRT-PCR**. Total RNA was isolated from tissues or cells using TRIzol reagent (Invitrogen) according to the manufacturer's instructions and then treated with DNase I (Promega). Reverse transcription of 1 μg of total RNA was performed using Transcript First Strand Synthesis Supermix (TransGen Biotech, AT301) to obtain complementary DNA (cDNA). qRT-PCR was performed using a 7500 Fast Real-Time PCR System (Applied Biosystems) and SYBR Green Supermix (TaKaRa) as described by the manufacturers. Raw data were normalized to GAPDH as the internal control and are presented as the relative expression levels, which were calculated as relative quantification (RQ) = $2^{-\Delta\Delta Ct}$. All gene-specific primers for qRT-PCR are listed in Supplementary Table 2.

**Western blot analysis**. Total protein was extracted from the cells with protein lysis buffer, and the protein concentration was subsequently measured by Lowry protein assay. Twenty micrograms of total protein from each sample was resolved by 10% or 12% sodium dodecyl sulfate-polyacrylamide gel electrophoresis (SDS-PAGE), transferred to a polyvinylidene difluoride membrane (Millipore), blocked in 5% nonfat milk and immunoblotted with various primary antibodies and horseradish peroxidase-conjugated secondary antibodies. Anti-phospho ERK1/2 (9101, 1:1000), anti-ERK1/2 (9102, 1:500), anti-phospho RelA (3033, 1:1000), anti-phospho p38 (9211, 1:500), and anti-p38 antibodies (9212, 1:500) were purchased from Cell Signaling Technology. An anti-cyclin E1 (11554-1-AP, 1:500) antibody was purchased from Proteintech. An anti-GAPDH (KC-5G5, 1:5000) antibody was purchased from Kangchen. An anti-actin antibody (A5441, 1:2000) was purchased from Sigma-Aldrich. Antibody binding was detected using SuperSignal West Pico Chemiluminescent Substrate (Pierce).

**Calcium flux**. Splenic B cells were incubated in 1640 medium with 10% FBS and the fluorescent $Ca^{2+}$ indicator Fluo-4-AM ester (2.5 mg/ml; Invitrogen) for 30 min at 37 °C, and the cells were then washed with medium and incubated for an additional 30 min at 37 °C. After addition of anti-IgM (10 μg/ml) or ionomycin (5 μM; Sigma-Aldrich), fluorescence kinetics were monitored every 5 s for 6 min with a Multimode Reader (excitation: 488 nm, emission: 525 nm) to reflect the calcium flux.

**Immunizations and ELISA**. To evaluate antigen-specific antibody responses, age-matched wild-type control, and iron-deficient mice (8-weeks-old) were injected intraperitoneally with 100 μg of DNP-KLH (TD antigen; Merck/Calbiochem) emulsified in 300 μl of complete Freund's adjuvant (CFA), 50 μg of TNP-LPS (TI-1 antigen; Sigma-Aldrich) or 25 μg of DNP-Ficoll (TI-2 antigen; Biosearch Technologies) in 300 μl of PBS. Blood serum was collected at the indicated times after immunization, and the concentrations of DNP- and TNP-specific antibody titers were assessed by ELISA (Southern Biotechnology Associates) following the manufacturer's instructions. To study germinal center formation, mice were intraperitoneally immunized with 5% SRBCs in 400 μl of PBS or with 100 μg of DNP-KLH in 300 μl of CFA. After 10 days, germinal center formation in the spleen was evaluated by flow cytometry after staining with PerCP-conjugated anti-B220 (553093, 1:200), FITC-conjugated anti-GL7(562080, 1:200), PE-conjugated anti-Fas (554258, 1:200), APC-conjugated anti-IgD (560868, 1:200) antibodies, APC-conjugated anti-CD21(558658, 1:200), FITC-conjugated anti-CD23(553138, 1:200) and PE-conjugated anti-IgM(553409, 1:200) were purchased from BD Biosciences. Alexa Fluor 488-conjugated rat anti-IgD (562023, BD Pharmingen, 1:100), biotin-conjugated peanut agglutinin (PNA, BA-0074, Vector Laboratories, 1:100), and Alexa Fluor 555-conjugated streptavidin (S32355, Invitrogen, 1:400). Vitamin D was checked using 25(OH) Vitamin D ELISA kit (Abcam) according to manufacturer's protocol.

**Immunohistochemistry**. Ten days after the mice were immunized with SRBCs, the spleens of iron-deficient mice and control mice were collected and embedded in optimum cutting temperature compound[38] and frozen at −80 °C. The sections were mounted (8 μm in thickness), fixed for 10 min in cold acetone, washed in PBS, blocked for 30 min with a blocking solution (PBS + 0.5% (wt/vol) BSA), and stained overnight with primary antibodies. The sections were then washed with PBS containing 0.5% (vol/vol) Tween and incubated with secondary antibodies for 2 h. The antibodies included Alexa Fluor 488-conjugated rat anti-IgD (BD Pharmingen), biotin-conjugated peanut agglutinin (PNA, Vector Laboratories), and Alexa Fluor 555-conjugated streptavidin (Invitrogen). Finally, the sections were washed and covered in GVAMount (Zymed Laboratories), and images were acquired using an Axio Observer.A1 microscope (Zeiss).

**ChIP assay**. After stimulation with anti-IgM (10 μg/ml) or LPS (2 μg/ml) for 24 h or 48 h, splenic B cells were cross-linked with 1% fresh formaldehyde for 10 min at room temperature, neutralized with glycine for 5 min and lysed in SDS lysis buffer. The cross-linked DNA was then sonicated and sheared into fragments 200–1000 base pairs in length with UCD-300 (Bioruptor). ChIP experiments were performed using a Chromatin Immunoprecipitation Kit (Millipore) according to the manufacturer's instructions to obtain ChIP-enriched DNA. For the ChIP-Seq assay, the 75-nt sequence reads generated by Illumina sequencing were mapped to the genome using the Burrows-Wheeler Aligner algorithm with default settings. Only reads that passed Illuminas purity filter, aligned with no >2 mismatches, and mapped uniquely to the genome were used in the subsequent analysis. Average of peak values of all active regions in the gene and within the gene margin were used to calculate the differentially enriched genes. For the ChIP-qPCR assay, subsequent qRT-PCR was performed to quantify the ChIP-enriched DNA. The data were normalized to the input. The antibodies for ChIP, including anti-H3K9me2 (39239, 1:100, 10 μg per ChIP), anti-H3K9me3 (39161, 1:100, 10 μg per ChIP) were purchased from Active Motif. The primers used for ChIP-qRT-PCR are listed in Supplementary Table 2.

**Statistical analysis**. All experiments were performed using 4–6 mice or at least two independent repeated experiments with cells. Unless otherwise indicated, the data in the figures are presented as the mean ± SEM unless otherwise indicated. Statistical analyses were performed using R (http://www.r-project.org/), and statistical significance was determined by two-tailed Student's $t$-test or Pearson's correlation coefficients test. For the population-based study, we used age and gender as stratified information to analyze and correlate iron metabolism indicators (ferritin, Serum Iron, UIBC, Tf Sat.) and antibody response (MV antibody titers) in all and various regions. The Wilcoxon rank sum test (also called the Mann–Whitney $U$-test) was used to analyze MV-specific antibody titers. Pearson's correlation test was used to determine the correlation coefficient $r$ and serum ferritin and MV concentration values were log10 transformed before analysis. For all statistical tests, $P < 0.05$ was considered significant ($*P < 0.05$, $**P < 0.01$).

**Reporting summary**. Further information on research design is available in the Nature Research Reporting Summary linked to this article.

## Data availability

The data that support the findings of this study are available within the article or its Supplementary Information files, and from the corresponding author upon reasonable request. The RNA-seq and ChIP-seq data that support the findings of this study have been deposited in Gene Expression Omnibus (NCBI) and are accessible through GEO Series accession numbers GSE77306 and GSE131716. A source data file is provided as a Supplementary Dataset. A reporting summary for this Article is available as a Supplementary Information file.

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

## Acknowledgements

We would like to thank Y Xiang, P. Zhou, X. Miu, and X. Guo for their technical assistance, H. Wei, B. Fu, D. Li, H. Wang, and J. Min for their suggestions, B. Peng, B. Qian, and Q. Jing for their support. This study was supported by grants from the National Basic Research Program (2014CB541904, 2014CB943600), the National Natural Science Foundation of China (31570902, 31370881, and 31530034), National Key Research and Development Program of China (2018YFA0507802).

## Author contributions

X.Z., Y.J. and F.W. conceived the project; Y.J. and X.Z. designed the research, analyzed the data, prepared the figures, and wrote the manuscript; Y.J., C.L., X.C. performed the experiments, L.H., Q.N.W., P.A., F.Z., C.C., S.Z., H.H., S.X. and F.W. examined B-cell percentage, antibody production, and analyzed data in humans; S.L., Y.S., H.L., Z.L., Y.T. and X.S. helped perform the experiments; Y.Z., Y. Hu., Q.I.W., D.Y., J.Z. and R.C. performed the ELISA experiments, L.M. and G.W. performed the ChIP assay and analyze the data; J.W. performed the ICP-MS; Y.L., G.W., S.Z., Y.W., Y.S., Y.E.C. and Y. Hao. oversaw the studies and engaged in discussions.

## Additional information

**Competing interests:** The authors declare no competing interests.

