## [Peer Review File · Nature Communications]

Reviewers' comments:

Reviewer #1 (Remarks to the Author):

The manuscript entitled "Iron-dependent Histone 3 Lysine 9 Demethylation Controls B Cell Proliferation and Humoral Immune Responses" by Jiang et al., assesses and demonstrates the important role of iron for B cell proliferation and function (production of antibodies) in a mouse model and reveals the possible mechanism behind these findings (i.e., the importance of demethylation of cyclin E1 promoter via KDM2B, KDM3B and KDM4C, leading to cyclin E1 induction and B cell proliferation). The authors also suggest that iron deficiency is leading to diminished humoral immunity/antibody response to measles vaccination by examining the correlations between serum iron/transferrin saturation and antibody titers following measles vaccination. The work, in particular the mouse studies, are technically sound and the presented results and conclusions are valid. However, I have a hard time linking these studies with the results in humans, and the claims that iron deficiency and similar mechanisms may be valid in humans as well. The authors need to modify the paper with a focus on the mouse model, or to provide more data for the human part of the study (please, see below).

MAJOR POINTS

1. The authors show data from a population-based study in 327 children after measles vaccination and demonstrate positive correlations between iron deficiency and measles antibody titers. While this may be true, not enough details are given, so that the readers can appropriately judge the presented results. For example;
 - a. Iron deficiency (the most common food deficiently) may be coexistent with other deficiencies. The authors need to give more details or at least comment on this point in the Discussion. Were vit. A, vit. D or other factors measured in this cohort?
 - b. It is not clear how many doses of vaccine the subjects had? The Methods and results also mention that subjects known to have measles were excluded. This is not enough – just including in the analysis vaccinated subjects from an area with measles outbreaks could affect the results, as vaccinated subjects can be protected from disease and still have a boost in immunity (higher antibody titers) from contact with wild type isolates without clinical symptoms.
 - c. The authors need to include more details on the statistical analysis, in particular for the population-based study. Measles vaccine-induced antibody titers can be associated with many factors, such as age, sex, time since last vaccination to enrollment in the study, living in an area with recent measles outbreaks etc. All these factors can be confounding variables in the analysis. Were these (or others factors) taken into consideration and adjusted for in the analysis? For example it is known that iron deficiency is often exacerbated by other factors including infections (e.g., HIV, TB, parasitic infections).
 - d. Since no data is presented about the time since vaccination for the study cohort, it is hard to imagine (for example) that iron deficiency measured years after vaccination is linked to the vaccine-induced immunity years earlier?
 - e. While the link between iron deficiency in humans and humoral immunity is likely, to strengthen the paper – the authors need to present for example:
 - data for the positive correlation between serum iron/transferrin saturation and antibody levels in humans after other vaccines (e.g., rubella, mumps, TT, Diphtheria etc.). Since the data presented from the mouse models points that the link between antibody titers and iron deficiency is likely a general phenomenon, more data on humans will be a plus. Were these subjects vaccinated with other vaccines?
 - some data in human B cells (from iron deficient subjects vs. iron normal) similar to the mouse studies.
2. Results, altered Ig responses in iron-deficient mice. The authors claim that the basal Ig levels between iron deficient and normal mice were not very different (only slightly reduced IgM levels), but the Supplementary figure 1d shows that this difference is statistically significant. The authors present antibody data in iron-deficient mice vs. normal mice after immunization with SRBC and KLH. Were other immunizations tried as well (e.g., components of the measles-mumps-

rubella vaccine or other vaccines)?

In addition, did the authors follow the antibody titers in iron-deficient mice vs. normal mice after immunization long-term, i/e, beyond 28 days. If the waning of immunity in the iron-deficient mice vs. normal mice is different, perhaps this result will be more relevant to human studies with iron and antibody measured years after immunization.

3. Do the authors have data from the mouse studies on the effect of iron deficiency on the memory B cell compartment of the long-lived plasma cell compartment after immunization in the mouse model (or the measles-specific memory B cells in the iron-deficient vs. iron-normal normal human subjects)?

4. While it is clear that the focus of the paper is on the B cell compartment (and humoral immunity) – more data (from the mouse model) on the effect of iron deficiency on T cell activation/function and perhaps on the innate cells will strengthen the paper as such effects are reported in the literature (this can be included in the Supplemental material).

The discussion should include more comments on the effects of iron deficiency on other cell types (in particular cell types that can be linked to humoral immunity).

5. Results, Iron is not required for the early BCR signaling. The authors present data in a mouse model that BCR or TLR early signaling is not affected by early signaling, but required for the middle and end stages? Is iron required for processes related to the generation of memory B cells or long-lived plasma cells. Perhaps more comments can be included in the Discussion.

6. Results, Iron is responsible for S entry and cyclin E1 induction during B cell proliferation. Please include data and/or comments in the Discussion on cyclin E1 induction and proliferation of other cell types.

7. Discussion. The paper needs more in-depth discussion on the presented results.

MINOR POINTS

1. Fig 1 and everywhere in the manuscript, define MV antibody titers mIU/mL, as generally accepted.

2. Fig.1 Low antibody group is defined as MV titer between 200 and 800 mIU/mL. I would argue that titers above 200 mIU/mL (and much higher 500 – 800 mIU/mL) are suggested to be associated with protection from disease and cannot be considered as low. More sound will be a comparison between normal/high measles vaccine-induced titers and titers below 200 mIU/mL (the threshold for protection).

3. Fig.1d – define the units/designations on the x axis and y axis. For example instead of $\log_{10}(\text{MV})$, use $\log_{10}(\text{MV Ab titer in mIU/mL})$.

4. Please, define all abbreviations in the manuscript (including in the figure legends). Define all units/designations on the x axis and y axis, and on the figure legends.

5. Fig 2 f and g – Were these representative of more than one experiment?

6. Supplementary fig. 2a. Is this STEAP3 protein expression or gene expression? Not very big difference is observed between the different cell types?

7. I would recommend that the manuscript is edited by a native English speaker to make the writing more fluent and to convey clearly the content.

Reviewer #2 (Remarks to the Author):

The paper from Jiang et al. starts with an interesting human clinical finding: an impaired measles vaccine response in human subjects with an iron deficit. Based on this altered immunoglobulin response in human they went further and tried to reproduce this finding in mice. The comparison between normal and iron-deprived mice gave similar results with impaired T-dependent and T-independent immune responses with a decreased number of germinal center B cells associated to a decrease of germinal center at the histological spleen level. They next used Steap3^{-/-} mice as controls since STEAP3 is required for iron uptake and highly expressed in B cells. No details are

given concerning the phenotype of these mice but the B cell development and maturation seemed not affected. At this point of the paper the authors focused their work on the proliferation defect that they found in these murine B cells. Since the differentiation of B cells to plasma cells required firstly a massive B proliferation they naturally found a decreased in the number of generated plasma cells after appropriate B cell stimulations.

This first part of the work present convincing data, with a high number of controls and the authors completed their work with data on the BCR and TLR signaling showing that both were not affected by iron deprivation. To complete this part I would suggest two additional explorations: *) to immunize their steap-/- mice and explore B cell response and germinal center formation in the spleen, **) to test and differentiate in vitro human peripheral blood B cells from subjects with low iron serum levels compared to normal subjects. In a recent paper from Liu et al. (Blood 2016; 127: 1067) they found a high prevalence of STEAP3 mutations in southern China and authors concluded that the deleterious effect in humans of these common abnormalities remains to be confirmed. The question would be, a guest at this point, if these people present an immune B defect...

The second part of the work is build and centered on the question of iron involvement in the B cell proliferation. The task is not easy but I was not convince by the data even though cyclin E1 seemed to be particularly lowered when cell are starved for iron. The experiments used almost exclusively culture conditions with deferoxamine, an iron chelator. We would need some data based on murine B cells collected in mice iron-restricted as in figure 2. Controls are missing and especially for transfected cells with sh-RNAs or with cyclin-E1 producing vectors (qPCR, at least). We also need larger insights at the transcriptional level, with multiple cell comparisons using RNA-seq approaches that need to be next combined with the ChIP-seq histone marks. JmjC KDMs members of the 2-OG-ferrous ion-dependent oxygenase are important in multiple biological processes, including, but not limited to, transcriptional regulation. We need therefore broader insights and at least two different drugs should be tested to confirm the H3K9me data. The drug IOX-1 as a broad spectrum of 2-oxoglutarate oxygenase inhibition and therefore the specificity of the effects should be challenged. We would also see some data with B cells from mice starved for iron. We haven't see any technical details concerning the ChIP experiments in the method section. The enrichment for H3K9me2/3 seem particularly low and data of the figure 8d are specially weak and over-specified...

The authors started their introduction by saying that the trace element iron is essential in many fundamental metabolic processes in cells and organisms. We have the feeling that it is too reductive to limit the cell effect of iron in B lymphocytes and humoral immunity to the unique control of Cyclin E1 expression, even in the context of the B cell proliferation function. Again, the main clinical finding that supports this work is very interesting and I would suggest staying more on the human B-cell side to first confirm the lack of proliferation in subjects with low serum iron levels...

Reviewer #1 (point by point response):

The manuscript entitled “Iron-dependent Histone 3 Lysine 9 Demethylation Controls B Cell Proliferation and Humoral Immune Responses” by Jiang et al., assesses and demonstrates the important role of iron for B cell proliferation and function (production of antibodies) in a mouse model and reveals the possible mechanism behind these findings (i.e., the importance of demethylation of cyclin E1 promoter via KDM2B, KDM3B and KDM4C, leading to cyclin E1 induction and B cell proliferation). The authors also suggest that iron deficiency is leading to diminished humoral immunity/antibody response to measles vaccination by examining the correlations between serum iron/transferrin saturation and antibody titers following, measles vaccination. The work, in particular the mouse studies, are technically sound and the presented results and conclusions are valid. However, I have a hard time linking these studies with the results in humans, and the claims that iron deficiency and similar mechanisms may be valid in humans as well. The authors need to modify the paper with a focus on the mouse model, or to provide more data for the human part of the study (please, see below).

MAJOR POINTS

1. The authors show data from a population-based study in 327 children after measles vaccination and demonstrate positive correlations between iron deficiency and measles antibody titers. While this may be true, not enough details are given, so that the readers can appropriately judge the presented results. For example; a. Iron deficiency (the most common food deficiency) may be coexistent with other deficiencies. The authors need to give more details or at least comment on this point in the Discussion. Were vit.A, vit. D or other factors measured in this cohort?

Response :

We really appreciate reviewer #1's comments. We totally agree with reviewer #1 that we had better provide more evidences to support the positive correlations between iron deficiency and reduced measles antibody titers. It seems to be more complicated in human study. To assess the involvement of vitamins in humoral immunity we

examined Vitamin D in some of our samples. Due to limited sample volume we did not examine Vitamin A. Our results showed that in population 10 years and above in which positive correlations between iron deficiency and measles antibody titers were observed, MV-specific IgG antibody titers was not significantly correlated with Vitamin D level. There was also no obvious correlation between serum iron/ferritin and Vitamin D. These data indicated that iron deficiency (the most common food deficiently) may not be coexistent with Vitamin D deficiencies in these population.

Please see the figure below.

However, in younger children population less than 10-years old) in which positive correlations between iron deficiency and measles antibody titers were not observed, we indeed observed a positive correlation between antibody levels and Vitamin D levels. Moreover, serum iron levels were negatively correlated with vitamin D in these population under 10 years of age.

Although it has been reported that Vitamin D supplementation may enhance humoral immune function and increase immunoglobulin levels, but our data suggested that

the positive correlation between Vitamin D and antibody deficiency only occurred in the younger children under 10-years old, not in the population above 10-years old. Our data suggested that iron deficiency and Vitamin D deficiency might resulted in humoral immune deficiency in the population above 10-years old and under year old, respectively. Since reduced antibody response in iron-deficient population is nor attributable to Vitamin D deficiency we did not add these data into the revised manuscript.

Furthermore, we also tested the content of other trace elements in the serum of some individuals with sufficient samples to assess the possibility of other deficiencies that may affect the MV response. We detected the content of other trace elements in serum samples by ICP-MS. Results indicated that in population aged 10 years and above, no significant correlation was observed between Magnesium, calcium, manganese, copper, zinc and MV response.

But interestingly, in younger children population less than 10-years old, there was a significant positive correlation between copper and MV-specific antibody titers, suggesting a potential role of copper ions in humoral immune response of children.

b. It is not clear how many doses of vaccine the subjects had? The Methods and results also mention that subjects known to have measles were excluded. This is not enough – just including in the analysis vaccinated subjects from an area with measles outbreaks could affect the results, as vaccinated subjects can be protected from disease and still have a boost in immunity (higher antibody titers) from contact with wild type isolates without clinical symptoms.

Response:

The population samples were taken from five different regions of Zhejiang Province, and measles in Zhejiang Province is at a low level, and these subjects also come from places where measles has no epidemic and very low incidents of disease in recent years, so as to avoid interference with disease infection. Measles vaccine is routine immunization of the population. In recent decades, Zhejiang Province routinely use two needles of measles vaccination for children aged 8 months and 18 months, with a dose of 0.5 ml. The source of the vaccine is China Biotechnology Co., Ltd. The information has been supplemented in the “Methods” section.

c. The authors need to include more details on the statistical analysis, in

particular for the population-based study. Measles vaccine-induced antibody titers can be associated with many factors, such as age, sex, time since last vaccination to enrollment in the study, living in an area with recent measles outbreaks etc. All these factors can be confounding variables in the analysis. Were these (or others factors) taken into consideration and adjusted for in the analysis? For example it is known that iron deficiency is often exacerbated by other factors including infections (e.g., HIV, TB, parasitic infections).

Response:

In the population-based study, people with infectious diseases such as HIV, TB and parasitic infections were excluded by reference to the standard of exclusion. In the analysis, we used age and gender as stratified information to analyze and correlate iron metabolism indicators (Serum Iron, UIBC, Tf Saturation) and antibody response (MV) in all and different regions. Due to the low incidence of measles in Zhejiang Province, individuals in this case were all not living in an area with recent measles outbreaks. We particularly focused on the differences in MV antibody response between the normal and iron-deficient groups and the differences in iron indicators between the normal and low MV antibody response groups in different regions, genders or age groups. More details on the statistical analysis for the population-based study were added in the “Methods” section.

d. Since no data is presented about the time since vaccination for the study cohort, it is hard to imagine (for example) that iron deficiency measured years after vaccination is linked to the vaccine-induced immunity years earlier?

Response:

According to the protocol described above, two needles of measles were immunized when people were aged 8 months and 18 months. In people of different ages, and measles vaccine-specific IgG antibody titers with iron index were checked at the same time in people of different ages so as to assess the correlation between vaccine response and iron index.

e. While the link between iron deficiency in humans and humoral immunity is likely, to strengthen the paper – the authors need to present for example:

-data for the positive correlation between serum iron/transferrin saturation and antibody levels in humans after other vaccines (e.g., rubella, mumps, TT, Diphtheria etc.). Since the data presented from the mouse models points that the link between antibody titers and iron deficiency is likely a general phenomenon, more data on humans will be a plus. Were these subjects vaccinated with other vaccines?

-some data in human B cells (from iron deficient subjects vs. iron normal) similar to the mouse studies.

Response:

Thanks for reviewer's suggestions. We tried to collect proper populations with other vaccines. But we failed to check serum iron/transferrin saturation and antibody levels in humans after other vaccines under current conditions.

We totally agree with the comments that we definitely need to check human B cells proliferation in iron deficient subjects. Given that after we isolated human B cells from iron deficient subjects, iron levels in B cells from iron-deficient patients would return to normal under normal culture conditions *in vitro* (rich iron ions in fetal bovine serum), we set up a new culture system to better simulate the proliferation of B cells *in vivo* in iron-deficient subjects and iron-normal subjects, in which we cultured B cells under 20 % serum from the same individuals that B cells were isolated from. We selected two iron-normal subjects and two iron-deficient subjects (Fig. 4g). For B cells from iron-deficient subjects cultured in iron-deficient serum (cultured in 20% autologous serum conditions), significantly fewer cells undergo division and proliferation under LPS-stimulated conditions compared with B cells from iron-normal subjects. What's more, for B cells from iron-deficient subjects, FAC supplementation could dramatically increase the proliferation of human B cells in response to TLR stimulation; and for B cells from iron-normal subjects, B cell proliferation was severely impaired after adding iron-chelating DFO in culture condition. These results further indicated a critical role of iron also in human B cell proliferation defects. Data were supplemented as Figure 4f-4i.

2.Results, altered Ig responses in iron-deficient mice. The authors claim that the basal Ig levels between iron deficient and normal mice were not very different (only slightly reduced IgM levels), but the Supplementary figure 1d shows that this difference is statistically significant.

Response:

Sorry, we did not claim very clearly. We have made correction about this in the revised manuscript.

The authors present antibody data in iron-deficient mice vs. normal mice after immunization with SRBC and KLH. Were other immunizations tried as well (e.g., components of the measles-mumps-rubella vaccine or other vaccines)?

Response:

We didn't try components of the measles-mumps-rubella vaccine in mice mode, but we used several immune antigens that are common in mouse models, including Polyclonal antigen SRBC, T cell-dependent antigen DNP-KLH, T cell-independent TI-1 antigen TNP-LPS and TI-2 antigen DNP-Ficoll. The immunization all showed similar humoral immunity defects in iron-deficient mice.

In addition, did the authors follow the antibody titers in iron-deficient mice vs. normal mice after immunization long-term, i/e, beyond 28 days. If the waning of immunity in the iron-deficient mice vs. normal mice is different, perhaps this result will be more relevant to human studies with iron and antibody measured years after immunization.

Response:

Thanks for great suggestions. In the previous experiments, mice were sacrificed after the last sample was taken within 1 month. In order to follow the antibody titers in iron-deficient mice vs. normal mice after long-term immunization, we collected eyelid blood samples and examined the antibody titers 2 months after the DNP-KLH immunization mice. Results showed significantly lower levels of antigen-specific IgG1 and IgM after long-term immunization in iron-deficient mice compared to control mice. Data were shown in Figure 2e.

3. Do the authors have data from the mouse studies on the effect of iron deficiency on the memory B cell compartment of the long-lived plasma cell compartment after immunization in the mouse model (or the measles-specific memory B cells in the iron-deficient vs. iron-normal normal human subjects)?

Response:

In DNP-KLH immunization experiments, according to the method reported in the literature, the mice were reimmunized for a second time at day 21, which lead to a

strong recall immune response at day 28 in normal mice, but in iron-deficiency mice, the recall immune response was weak. These results showed that the humoral immune function and antigen-specific immunoglobulin levels of the iron-deficient mice after re-immunization remain significant defect compared with control iron-sufficient mice and might suggested us an essential role of iron in memory B cell compartment. In this study we demonstrated the critical role of iron in B cell proliferation which is critical and essential for downstream events of B cells including germinal center formation, antibody production, memory B cell development. Generally, germinal center formation is essential for memory B cell development. We observed dramatically reduced germinal center formation, deficient primary and recall humoral immunity after antigen immunization in iron deficient mice. Since iron deficiency leads to impaired B cell proliferation, we did not check measles-specific memory B cells in iron-deficient mice. In order to access whether iron was related to the generation of memory B cells or long-lived plasma cells, we need to set up a new experimental system in which iron deficiency bypass deficient B cell proliferation.

4. While it is clear that the focus of the paper in on the B cell compartment (and humoral immunity) – more data (from the mouse model) on the effect of iron deficiency on T cell activation/function and perhaps on the innate cells will strengthen the paper as such effects are reported in the literature (this can be included in the Supplemental material).

The discussion should include more comments on the effects of iron deficiency on other cell types (in particular cell types that can be linked to humoral immunity).

Response:

We highly appreciate great suggestions. We think that the point is very important. Certainly, iron deficiency may impair the functions of other types of cells including T cells that are also indispensable for antibody production, particularly for TD antigens. Since histone demethylation critically functions in T cell differentiation and function (Zhu et al., 2010). We also measured proliferative responses of T cells to TCR stimulation in different concentrations of DFO, as it is shown in Figure S4b, iron deprivation also leads to impaired T cell proliferation, however, T cell proliferation is

significantly less dependent on iron than B cell proliferation, as assessed by the IC50. These suggested that in different types of cells in which rapid proliferation upon activation is essential for other events, the dependence on iron is different. Based on our data iron has no obvious effects on the survival and differentiation of both T cells and B cells. Therefore, we believe that iron have a certain selectivity for the proliferation and function of B cells. We have discussed these issues and added the data in supplementary figures.

5. Results, Iron is not required for the early BCR signaling. The authors present data in a mouse model that BCR or TLR early signaling is not affected by early signaling, but required for the middle and end stages? Is iron required for processes related to the generation of memory B cells or long-lived plasma cells. Perhaps more comments can be included in the Discussion.

Response:

We lack direct evidence if iron is required for the generation of memory B cells or long-lived plasma cells. But the re-immunization experiment mentioned above suggested that memory B cell function might also be defect in iron-deficient mice. We have included more comments in the “Discussion” section.

6.Results, Iron is responsible for S entry and cyclin E1 induction during B cell proliferation. Please include data and/or comments in the Discussion on cyclin E1 induction and proliferation of other cell types.

Response:

Thanks for your excellent suggestions. We have added data from T cells on cyclin E1 induction (Figure S4c) and proliferation to the supplementary figure and corresponding comments to the “Discussion” section.

7. Discussion. The paper needs more in-depth discussion on the presented results.

Response:

Thank you for your kind suggestion, we have added more in-depth discussion as you can see in the revised manuscript.

MINOR POINTS

1. Fig 1 and everywhere in the manuscript, define MV antibody titers mIU/mL, as generally accepted.

Response:

Thank you for your kind suggestion, we have changed the unit of MV antibody titers to mIU/mL.

2. Fig.1 Low antibody group is defined as MV titer between 200 and 800 mIU/mL. I would argue that titers above 200 mIU/mL (and much higher 500 – 800 mIU/mL) are suggested to be associated with protection from disease and cannot be considered as low. More sound will be a comparison between normal/high measles vaccine-induced titers and titers below 200 mIU/mL (the threshold for protection).

Response:

Thanks for kind suggestions. Yes, we have added the comparison between normal/high measles vaccine-induced titers and titers below 200 mIU/mL.

3. Fig.1d – define the units/designations on the x axis and y axis. For example instead of $\log_{10}(MV)$, use $\log_{10}(MV \text{ Ab titer in mIU/mL})$.

Response:

Thanks for suggestions. Yes, we have added the units/designations on the x axis and y axis as you have pointed out.

4. Please, define all abbreviations in the manuscript (including in the figure legends). Define all units/designations on the x axis and y axis, and or the figure legends.

Response:

Thanks for suggestions. We have defined all abbreviations all units/designations on the x axis and y axis in the whole manuscript, including the figure legends.

5. Fig 2 f and g – Were these representative of more than one experiment?

Response:

Thanks for your mention. Fig 2f and 2g were definitely representative of more than one experiment, Fig 2g were statistical results for Fig 2f, and 3 mice were checked for each experiment. We have made it more clear in figure legends.

6. Supplementary fig. 2a. Is this STEAP3 protein expression or gene expression? Not very big difference is observed between the different cell types?

Response:

Supplementary fig. 2a (now Fig. S3a) showed the STEAP3 gene expression. Although we did not observe a big difference in STEAP3 expression in the different cell types, the STEAP3 gene expression in B cells is significantly higher than that of other members of STEAP family.

7. I would recommend that the manuscript is edited by a native English speaker to make the writing more fluent and to convey clearly the content.

Response:

Thank you for your recommendation. With the help of NPG language editing service, we have re-edited the whole manuscript.

Reviewer #2 (point by point response):

The paper from Jiang et al. starts with an interesting human clinical finding: an impaired measles vaccine response in human subjects with an iron deficit. Based on this altered immunoglobulin response in human they went further and tried to reproduce this finding in mice. The comparison between normal and iron-deprived mice gave similar results with impaired T-dependent and T-independent immune responses with a decreased number of germinal center B cells associated to a decrease of germinal center at the histological spleen level. They next used *Steap3*^{-/-} mice as controls since STEAP3 is required for iron uptake and highly expressed in B cells. No details are given concerning the phenotype of these mice but the B cell development and maturation seemed not affected. At this point of the paper the authors focused their work on the proliferation defect that they found in these murine B cells. Since the differentiation of B cells to plasma cells required firstly a massive B proliferation they naturally found a decreased in the number of generated plasma cells after appropriate B cell stimulations.

This first part of the work present convincing data, with a high number of controls and the authors completed their work with data on the BCR and TLR signaling showing that both were not affected by iron deprivation. To complete this part I would suggest two additional explorations: *) to immunize their *steap*^{-/-} mice and explore B cell response and germinal center formation in the spleen,

Response:

We greatly appreciate Reviewer#2's comments and suggestions. In order to test the role of STEAP3 in B cell response we tried to immunize *Steap3*^{-/-} mice and wild-type mice with T cell-independent (TI) antigen 2,4,6-trinitrophenyl (TNP)-LPS (that primarily lead to IgM and IgG3 antibody responses). Compared with wild-type mice, TI antigen-induced secretion of antigen-specific IgG3 and IgM was severely attenuated in *Steap3*^{-/-} mice. Data were added as Figure 3d.

But unfortunately, the *Steap3*^{-/-} mice used in our experiment are conventional knockout mice. Up to now we haven't got conditional knockout mice of *Steap3*. We found *Steap3* conventional KO mice were abnormal and showed severe anemia. The architecture of spleens in *Steap3* conventional KO mice possibly due to altered macrophage functions which will interfere with the conclusions about the effects of *Steap3* on B cell functions. Thus, we think that *Steap3* conventional KO mice are not very suitable for antigen immunoassay. To investigate whether B cell functions including the development, proliferation and survival of B cells in *Steap3*^{-/-} mice were intrinsic to the B cells, extrinsic to stromal cells in the microenvironment of spleens, we generated bone marrow chimeric mice by co-transferring the bone marrow cells from CD45.1 wild-type and CD45.2 *Steap3*^{-/-} mice. The chimeric mice appeared normal. And B cells consisting of wild-type and *Steap3*^{-/-} B cells from splenocytes of the same mice were isolated and stimulated with LPS or anti-IgM to assess role of *Steap3* in B cell function. *Steap3*^{-/-} CD45.2 B cells proliferated poorly in response to both TLR and BCR stimulation in the same condition compared with wild-type CD45.1 B cells. We considered that these results may better demonstrate the effect of *Steap3* on B cell function.

****) to test and differentiate in vitro human peripheral blood B cells from subjects with low iron serum levels compared to normal subjects.**

Response:

Thanks for the great suggestions. Similar as Reviewer #1's questions. Considering that iron levels in B cells from iron-deficient patients would return to normal level under normal culture conditions *in vitro* (rich iron ions in fetal bovine serum and medium), in order to better simulate the proliferation of B cells in iron-deficient

subjects and iron-normal subjects, we selected two iron-normal subjects and two iron-deficient subjects, retained serum during the process of isolating human B cells from peripheral blood seeking to culture B cells under autologous serum conditions. For B cells from iron-deficient subjects cultured in iron-deficient serum (cultured in 20% autologous serum conditions), significantly fewer cells undergo division and proliferation under LPS-stimulated conditions compared with B cells from iron-normal subjects. What's more, for B cells from iron-deficient subjects, FAC supplementation could dramatically increase the proliferation of human B cells in response to TLR stimulation; and for B cells from iron-normal subjects, B cell proliferation was severely impaired after adding iron-chelating DFO in culture condition. These results further indicated a critical role of iron also in human B cell proliferation defects. Data were supplemented as Figure 4f-4i.

In a recent paper from Liu et al. (Blood 2016; 127: 1067) they found a high prevalence of STEAP3 mutations in southern China and authors concluded that the deleterious effect in humans of these common abnormalities remains to be confirmed. The question would be, a guest at this point, if these people present an immune B defect...

Response:

We contacted the corresponding author professor Xu for helping us to reach some patients with loss-of-function *Steap3* mutations, seeking to detect B cell function in these patients, but unfortunately, we failed in persuading patients to participate and cooperate in our research. For this question, in Jabara et al. paper from Nature Genetics reported a missense mutation in transferrin receptor 1 (TfR1) in patients caused a combined immunodeficiency characterized by impaired proliferation and function of T and B cells, and STEAP3 (associates with TfR1) could partially rescues transferrin uptake in patient-derived fibroblasts, suggesting that STEAP3 may be involved in the function of B cells.

The second part of the work is build and centered on the question of iron involvement in the B cell proliferation. The task is not easy but I was not convince by the data even though cyclin E1 seemed to be particularly lowered when cell are starved for iron. The experiments used almost exclusively culture conditions with deferoxamine, an iron chelator. We would need some data based on murine B cells collected in mice iron-restricted as in figure 2.

Response:

We highly appreciate the suggestions. We performed the experiment as Reviewer# 2 suggested. We isolated splenic B cells from control and iron-deficient mice immunized with T cell-dependent antigen DNP-KLH and T cell-independent TI-1 antigen TNP-LPS, and checked the cyclin E1 expression by qPCR. Data indicated that cyclin E1 induction was significantly decreased in iron-deficient mice, which further support results of our conclusion based on *in vitro* experiments. Data were supplemented as Figure 6h.

Controls are missing and especially for transfected cells with sh-RNAs or with cyclin-E1 producing vectors (qPCR, at least). We also need larger insights at the transcriptional level, with multiple cell comparisons using RNA-seq approaches that need to be next combined with the ChIP-seq histone marks. JmjC KDMs members of the 2-OG-ferrous ion-dependent oxygenase are important in multiple biological processes, including, but not limited to, transcriptional regulation. We need therefore broader insights and at least two different drugs should be tested to confirm the H3K9me data. The drug IOX-1 as a broad spectrum of 2-oxoglutarate oxygenase inhibition and therefore the specificity of the effects should be challenged. We would also see some data with B cells from mice starved for iron.

Response:

Thank you for the careful comments. We have supplemented the cyclin E1 expression of B cells infected with sh-RNAs (Figure S6f) lentivirus or with cyclin E1 producing lentivirus (Figure 6g).

To get larger insights at the transcriptional level and epigenetic changes, we combined RNA-seq data analysis together with ChIP-seq results. When comparing ChIP-Seq intervals in control and iron-deficient B cells, 1559 genes displayed more

than 2-fold upregulation of H3K9me2 modifications upon iron deprivation. In line with the notion that H3K9me2 modification is critical for gene transcription activation, the difference in the peaks between control and iron-deficient B cells was more obvious within the promoter areas and throughout gene bodies. To further correlate epigenetic changes with direct gene regulation, we noticed that a total of 127 genes might be directly regulated by iron-dependent JmjC demethylases, as they showed an increased H3K9me2 modification and reduced expression levels in the absence of iron. Data has been shown in Figure 7c.

As Reviewer #2 pointed out, we totally agree that the specificity of the IOX-1 effects might be challenged. At present, IOX1 was the only commercially available inhibitor of 2-OG. On the other hand, we can only use inhibitor of 2-OG, IOX1, to provide another evidence, together with iron deficiency, suggesting the effect of JmjC enzyme in B cell proliferation since both 2-OG and iron are essential for JmjC enzyme. These prompted us to further find which JmjC enzymes are responsible for B cell proliferation.

We haven't see any technical details concerning the ChIP experiments in the method section. The enrichment for H3K9me2/3 seem particularly low and data of the figure 8d are specially weak and over-speculated...

Response:

Thanks for reviewer's mention. We have added the technical details of the ChIP experiments in the "Methods" section. The question "the enrichment for H3K9me2/3 seem particularly low" is very important and professional. We had an in-depth

discussion with the people performed the sequencing and bioinformatics sections of the study. We speculated the reason was that the total raw reads were only 20 M when we performed ChIP-seq. Ideally, the raw reads should have been terminated after at 50M. Fortunately, with ChIP-qPCR assay we confirmed that the enrichment for H3K9me2/3 at multiple sites of cell cycle-related genes.

For figure 8d we agree that the data evidence is not strong enough and is a bit over-speculated, we have adjusted our statements to a more rigorous way. Thanks for reviewer's comments.

The authors started their introduction by saying that the trace element iron is essential in many fundamental metabolic processes in cells and organisms. We have the feeling that it is too reductive to limit the cell effect of iron in B lymphocytes and humoral immunity to the unique control of Cyclin E1 expression, even in the context of the B cell proliferation function. Again, the main clinical finding that supports this work is very interesting and I would suggest staying more on the human B-cell side to first confirm the lack of proliferation in subjects with low serum iron levels...

Response:

We greatly appreciate Review#2's positive comments and totally agree his/her suggestions. We tried to find the population with steap3 mutation to perform more experiments, examine the effects of iron on human B cell proliferation and explore the related epigenetic regulation. Unfortunately, we have no steap3 mutated persons available right now. We hope we can continue this project to perform more experiments on human B cells in the future.

REVIEWERS' COMMENTS:

Reviewer #1 (Remarks to the Author):

Overall the authors considered in depth all comments and addressed them properly. This improved the quality and completeness of the revised manuscript.

Reviewer #2 (Remarks to the Author):

The paper was significantly improved and answers to my comments are admissible. I propose to add to the discussion section the information concerning people with STEAP3 mutations, add also the Blood 2016 reference, and the fact that it would be of interest to explore further these subjects for B cell differentiation and response to vaccination.

REVIEWERS' COMMENTS:

Reviewer #1 (Remarks to the Author):

Overall the authors considered in depth all comments and addressed them properly. This improved the quality and completeness of the revised manuscript.

We appreciate so much for the time and effort the reviewer has put into your comments. This helped us a lot in improving the study and the whole paper.

Reviewer #2 (Remarks to the Author):

The paper was significantly improved and answers to my comments are admissible.

I propose to add to the discussion section the information concerning people with STEAP3 mutations, add also the Blood 2016 reference, and the fact that it would be of interest to explore further these subjects for B cell differentiation and response to vaccination.

Thank you so much for the time and effort you have put into your comments. We totally agree with this proposal and we have added the information concerning people with STEAP3 mutations to the discussion section as following:

In a study using large cohorts of normal individuals in China, Xu. et.al found high prevalence of STEAP3 mutation (5.3% in 2338 individuals)³¹. 16 different loss-of-function STEAP3 mutations were identified, which resulted in severely or moderately impaired ferrireductase activity³¹. But the deleterious effect in humans of these common abnormalities remains to be confirmed. Considering our study showed that Steap3-KO B cells from mice model exhibit severe defects in B cell proliferation and immune function, whether these people with STEAP3 mutations (especially those with loss-of-function mutations) present an immune B defect seems to be an interesting question worth further exploring.